# BCXN-PINN for Complex Geometry: Solving PDEs with Boundary Connectivity Loss

## Abstract

We present a novel loss formulation for efficient learning of complex dynamics from governing physics, typically described by partial differential equations (PDEs), using physics-informed neural networks (PINNs). In our experiments, existing versions of PINNs are seen to learn poorly in many problems, especially for complex geometries, as it becomes increasingly difficult to establish appropriate sampling strategy at the near boundary region. Overly dense sampling can adversely impede training convergence if the local gradient behaviors are too complex to be adequately modelled by PINNs. On the other hand, if the samples are too sparse, PINNs may over-fit the near boundary region, leading to incorrect solution. To prevent such issues, we propose a new Boundary Connectivity (BCXN) loss function which provides local structure approximation at the boundary. Our *BCXN-loss* can implicitly or explicitly impose such approximations during training, thus facilitating fast physics-informed learning across entire problem domains with order of magnitude fewer training samples. This method shows a few orders of magnitude smaller errors than existing methods in terms of the standard L2-norm metric, while using dramatically fewer training samples and iterations. Our proposed BCXN-PINN method does not pose any requirement on the differentiable property of the networks, and we demonstrate its benefits and ease of implementation on both multi-layer perceptron and convolutional neural network versions as commonly used in current physics-informed neural network literature.

## 1 Introduction

Physics-informed neural networks (PINNs) have emerged as a promising method for learning the solution of dynamical system from the governing physics (Raissi et al., 2019). PINNs have recently been studied for a wide range of physical phenomena and applications across science and engineering domains — electromagnetic, fluid dynamics, heat transfer, etc (Karniadakis et al., 2021; Cuomo et al., 2022). The distinctive feature of PINNs is the use of governing physics law, typically in the form of partial differential equations (PDEs), as the learning objective. This physics-informed learning constrains the PINN from violating the underlying physics at all training points sampled from the problem domain.

Existing PINNs evaluate the PDE constraints in their training loss by either automatic differentiation (AD) or numerical differentiation (ND)-type method (Wandel et al., 2020). While both methods have their pros and cons, ND-type PDE loss can be flexibly implemented across many different neural network (NN) architectures, including both multi-layer perceptrons (MLPs) and convolutional neural networks (CNNs), because they do not require the NN to retain differentiability, unlike AD. Recent studies (Gao et al., 2021; Fang, 2021; Chiu et al., 2022) have also suggested that *ND-loss* can more robustly and efficiently produce accurate solutions with fewer training samples, whereas conventional *AD-loss* are prone to failure during training . This is because ND-type methods approximate high order derivatives using *PINN output from neighbouring samples*, hence, they can effectively connect sparse samples into piecewise regions via these local approximations, thereby facilitating fast physics-informed learning across the entire domain with less dense samples.

When dealing with irregular geometries, it becomes increasingly difficult for existing PINNs to perfectly connect training samples in the domain's interior to the boundary. Failing to do so can cause undesirable training failure as the PINN starts to over-fit at the near boundary region. Since

many PDEs of practical interest are boundary-value problems, it is desirable to have the PINNs model the correct boundary behaviors. While adding dense samples to better refine the piecewise local regions near the boundary may improve accuracy, the extent to which sampling needs to be increased is empirical, and a denser sampling strategy may adversely impede training convergence.

Hence, we propose a loss function formulation in this work that helps provide an approximation to the local gradient behaviour at the boundary, thereby restoring connectivity between domain boundary and near boundary interior samples. This new boundary connectivity (*BCXN*)-loss function is key to a novel class of BCXN-PINN method which can more efficiently learn the solution to PDEs with fewer training samples, regardless of domain geometry; see example in Fig. 1. In addition, this method can be jointly implemented with other PINN advances such as in loss balancing, domain decomposition, adaptive sampling and other improved optimization methods (Zeng et al., 2022).

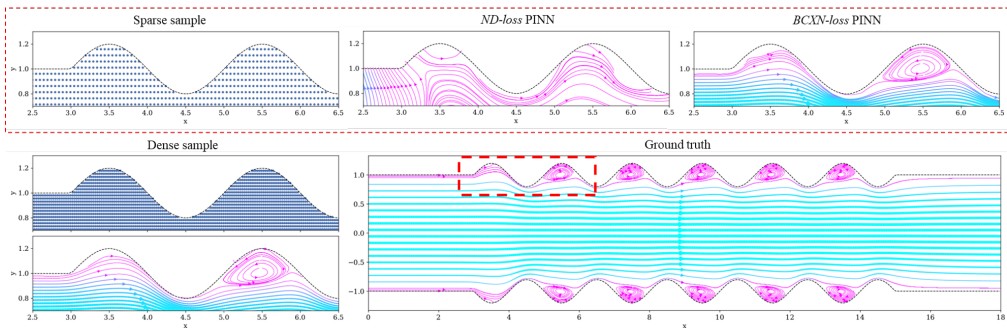

Figure 1: PINNs learning the solution of 2D N-S equations in a complex geometry (wavy channel flow problem, $Re = 100$). Our BCXN-PINN method can learn accurate solution with faster speed (50% less training iterations) and fewer (50% less) training samples.

In the rest of this work, we present two versions of this *BCXN-loss*: i) a soft forcing approach which imposes a linear approximation constraint via an additional loss term, and ii) a direct forcing approach which strongly enforces a linear constraint during the evaluation of PDE loss at near-boundary samples. With the latter approach, there is no longer a need to explicitly evaluate the boundary samples as the exact BCs have been implicitly "infused" into the near-boundary samples. Moreover, the direct forcing *BCXN-loss* can be beneficial to CNN-type architectures which utilize a structured grid and lack the ability to model exact domain boundaries for irregular geometries.

We present comprehensive experiments to demonstrate i) the flexible implementation of BCXN-PINN method for both MLP and CNN architectures; and ii) the effectiveness of BCXN-PINN for learning multiple complex fluid dynamical systems, spanning forward, inverse and meta-model problems in two-dimensions (2D) and three-dimensions (3D). Compared to conventional PINNs with the *AD-* and *ND-loss*, our BCXN-PINNs with *BCXN-loss* are shown to be capable of tackling challenging PDE problems while using fewer training samples, hence expanding the exciting potential of PINNs for learning complex dynamical evolutions encountered in the real-world.

## 2 RELATED WORK

**Efficient sampling in PINNs.** The theoretical limit of physics-informed loss learning in relation to training samples has been provided by prior studies (Lu et al., 2021c; Mishra & Molinaro, 2022). With the goal of improving the PINN training speed for practical applications (Markidis, 2021), several studies have focused on efficient sampling strategies such as importance sampling, adaptive sampling, and sequential sampling to reduce the amount of training samples being required during PINN trainings (Anitescu et al., 2019; McClenny & Braga-Neto, 2020; Wight & Zhao, 2020; Nabian et al., 2021; Lu et al., 2021a; Lye et al., 2021; Daw et al., 2022; Mattey & Ghosh, 2022; Wu et al., 2023). Domain decomposition and parallelization strategies have also been explored to speed up the training (Jagtap et al., 2020; Jagtap & Karniadakis, 2021; Shukla et al., 2021; Dong & Li, 2021; Li et al., 2019; Kharazmi et al., 2021). Our method differs from these works in that we make physics-informed learning more robust in the sparse sample regime via a newly-proposed *BCXN-loss*.

**CNN architecture and numerical differentiation (ND)-type loss for PINNs.** CNN-based formulation allows us to design and train larger, more powerful networks, hence it has potential to be scalable for more complex, large-scale PDE problem (Wandel et al., 2020; Gao et al., 2021; Wandel et al., 2021; Ranade et al., 2021; Wandel et al., 2022; Ren et al., 2022). However, cumbersome coordinate transformations may need to be performed to better handle the irregular domain. CNN-architecture PINNs usually utilize ND methods for computing the PDE loss on their input grid, which is much cheaper to compute as compared to AD-based loss (Gao et al., 2021). Hybrid frameworks that couple both AD- and ND-type loss have also been proposed for MLP architecture PINNs to unify the advantages of both methods (Jagtap et al., 2020; Kharazmi et al., 2021; Mitusch et al., 2021; Fang, 2021; Chiu et al., 2022). Our method further augments the efficiency and applicability of ND-type loss on irregular geometries.

**Enforcing BC constraint in PINN loss.** It is important for the PINNs to prioritize learning the correct boundary behaviors (Shin et al., 2020; Wang et al., 2022). The BCs are usually enforced into the PINN loss as a soft constraint using penalty method. Various strategies have been explored in this context to dynamically calibrate the relative important between PDE and BC constraints during PINN training (Elhamod et al., 2020; Bischof & Kraus, 2021; Jin et al., 2021; Thanasutives et al., 2021; Maddu et al., 2022; de Wolff et al., 2022; van der Meer et al., 2022; Xiang et al., 2022; Huang et al.; Wang et al., 2021a). There are also other approaches to bypass the loss balancing issue. For example, one can either devise an ansatz function such that the BCs are exactly satisfied by construction or implicitly formulate the BC constraints into the PDE loss (Lagaris et al., 1998; 2000; McFall & Mahan, 2009; Berg & Nyström, 2018; Nabian & Meidani, 2019; Karumuri et al., 2020; Wang & Zhang, 2020), leaving only single loss term from PDE residuals to be optimized. Another example is the use of augmented Lagrangian method to impose the BC constraints into PINN loss (Lu et al., 2021b). Our method implicitly incorporates BCs into the PDE loss via a different strategy.

## 3 PRELIMINARY

### 3.1 GOVERNING PHYSICS - INCOMPRESSIBLE NAVIER-STOKES (N-S) EQUATIONS

The present study focuses on learning fluid dynamics with PINNs. Fundamentally, PINNs have been shown to be applicable to many different physical systems. We consider fluid problems where the governing physics are the steady-state, incompressible N-S equations derived from the conservation of mass and momentum:

$$\nabla \cdot \vec{u} = 0 \tag{1a}$$

$$(\vec{u} \cdot \nabla)\vec{u} = Re^{-1}\Delta\vec{u} - \nabla p \tag{1b}$$

In the above PDEs, the primitive variables $\vec{u}$ and $p$ are velocity vector and pressure while Reynolds number ($Re$) represents the ratio of inertial to viscous forces. $\vec{u}$ consists of 2 components $(u, v)$ for a 2D case and 3 components $(u, v, w)$ for a 3D case. While Cartesian coordinates are used in this work, this formulation is extendable to other coordinate systems.

### 3.2 MLP- AND CNN-ARCHITECTURE PINNs

We use a fully connected DNN architecture (e.g., MLP, CNN) to represent the solution of the dynamical process $\boldsymbol{U} = [\vec{u}, p]^T$. For MLP architecture, the input $\vec{x}$ is a point coordinate in 2D or 3D spatial domain. For CNN architecture, the input $\boldsymbol{X}$ is a tensor with a fixed shape, comprising the entire (discretized) spatial domain. The accuracy of the PINN outputs $\boldsymbol{U}(\vec{x};\boldsymbol{w})$ given input $\vec{x}$ is determined by the network parameters $\boldsymbol{w}$, which are optimized w.r.t. the PINN loss function during training. The PINN loss function is defined as the composition of a PDE loss component ($L_{PDE}$) and a BC loss component ($L_{BC}$):

$$L_{PINN} = \lambda_{PDE}L_{PDE} + \lambda_{BC}L_{BC} \tag{2a}$$

$$L_{PDE} = \left\|\nabla \cdot \vec{u}(\vec{x};\boldsymbol{w})\right\|^2_\Omega + \left\|(\vec{u}(\vec{x};\boldsymbol{w}) \cdot \nabla)\vec{u}(\vec{x};\boldsymbol{w}) - Re^{-1}\Delta\vec{u}(\vec{x};\boldsymbol{w}) + \nabla p(\vec{x};\boldsymbol{w})\right\|^2_\Omega \tag{2b}$$

$$L_{BC} = \left\|B[u(\vec{x};\boldsymbol{w})] - \vec{u}(\vec{x})\right\|^2_{\partial\Omega} \tag{2c}$$

The PDE loss penalizes deviation from governing N-S equations for the PINN output $U(\vec{x};w)$ over the fluid domain $\vec{x} \in \Omega$, whereas the BC loss penalizes deviation from the desired Dirichlet boundary condition $\vec{u}_{BC}(\vec{x})$ which constrains the velocity field at the domain boundary $\vec{x} \in \partial\Omega$. The relative weights $\lambda_{PDE}$ and $\lambda_{BC}$ in Eq. (2) a control the trade-off between different loss components.

The PDE and BC loss components are defined over a continuous domain, but for practical reasons, we evaluate the discretized PINN loss from a finite set of $n$ samples $D = \{(\vec{x}_i)\}_{i=1}^n$ during training. The CNN-architecture PINNs naturally acquire an input with equidistantly spaced grid where all the samples are coming from. For the MLP-architecture PINNs, it is also convenient to design training samples based on equidistantly spaced grid, then $m(<n)$ samples can be randomly drawn to compute the PINN loss during each stochastic gradient descent (SGD) mini-batch training iteration.

Although it is ideal for a PINN to satisfy the PDE constraint on a very dense set of samples, i.e., $\Delta\vec{x} \to 0$ where $\Delta\vec{x}$ is the distance between two adjacent sample points, training such PINN may not be computationally feasible even with the state-of-the-art SGD variants. For a more complex physics problem, the training on dense samples is naturally more computationally expensive, and is also more likely to become unsuccessful due to adverse impact by some local region with very complex gradient behaviours, e.g., abnormally high gradient (Fuks & Tchelepi, 2020; Michoski et al., 2020; Ramabathiran & Ramachandran, 2021; Ji et al., 2021; Lucor et al., 2021; Huang et al.). Given the same problem, training PINNs on fewer samples means fewer restrictions (the PDE constraint that needs to be satisfied) and an increased likelihood of avoiding problematic gradient points, thereby accelerating training. The *relaxation of PDE constraint via reducing the sample density makes learning physically meaningful solution feasible*, although it may introduce some approximation error to the solution. Note that there is still an open question as to how to strike the best balance between sample density and solution accuracy. PINN methods that can consistently produce more accurate solutions with fewer training samples are nevertheless hugely advantageous for learning complex dynamical evolutions encountered in the real-world.

### 3.3 LEARNING PINNs ON FEWER SAMPLES

Conventional PINNs evaluate their PDE constraints with either AD or ND. On a simple 1D convection-diffusion equation problem as exemplified in Fig.2, PINNs with both *AD-* and *ND-loss* exhibit incorrect solution when sparse samples are used for training. Note that this is not the only possible failure mode for PINNs (Krishnapriyan et al., 2021). The issue is that although the training is fast and the convergence is good (training losses are close to machine precision, i.e., $< 1\text{e-}10$), their final solutions are incorrect. Moreover, their failure patterns are different.

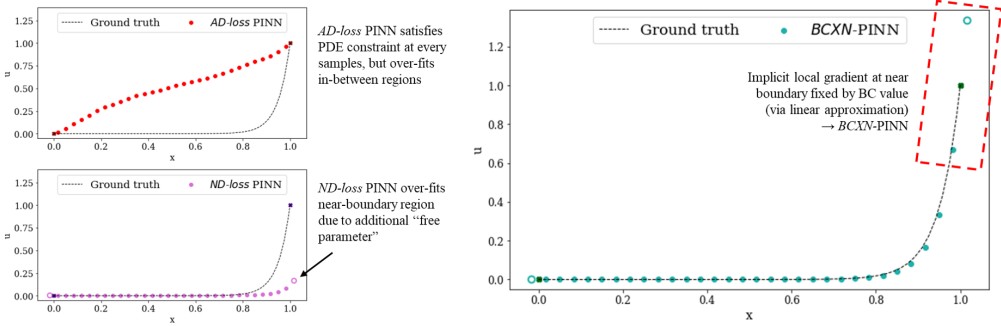

Figure 2: Schematic of different PINNs learning the solution of 1D convection-diffusion equation. For PINN with *ND-* and *BCXN-loss*, external stencil points (x-marked) are evaluated for computing the PDE constraint at near boundary samples. The ground truth solution is the dashed line.

**Failure pattern of *AD-loss* PINN.** AD has become a popular method for evaluation of the PDE constraint in PINN loss because it can exactly compute the derivative terms at any sample location directly from the input and the network weights. However, the *AD-loss* requires dense samples to guide the correct training. When samples are sparse relative to the local complexity, a highly flexible, over-parameterized PINN "over-fits" in between regions, leading to incorrect solution even as the PDE constraint is fulfilled at all sample points (Fig. 2 example).

**Failure pattern of *ND-loss* PINN.** On the contrary, *ND-loss* are more robust to sample density. The fundamental difference is that ND-type methods approximate the derivative terms by using *PINN output from neighbouring samples*, e.g., a finite difference-type stencil, which connects sparse samples into piecewise continuous regions to facilitate fast training across the entire problem domain. The use of lower order approximation to the derivative terms also provides certain structural bias for PINNs to be more easily trained. For example, abnormally high gradients are regularized so that it is easier for PINNs to learn the solution, with the accuracy being dependent on the approximation employed. Moreover, ND methods do not pose any requirement on differentiability of the network, and they naturally fit the CNN-architecture which utilizes a connected set of grid points as input.

The example in Fig. 2 demonstrates that *ND-loss* can still be subject to undesired training failure. The reason here is because the near-boundary training samples do not perfectly connect to the domain boundary through their stencils (as illustrated in Fig. 3). Instead, their stencils fall outside the boundary. Their outputs essentially act as free parameters during optimization for satisfying the PDE constraint at near-boundary samples, and the PINN training may "over-fit" the near-boundary region to more easily jointly satisfy the PDE and BC constraints. *This becomes a major issue for ND-loss when dealing with irregular geometries*, because it is almost impossible to design sample locations such that their stencils perfectly connect the domain boundary to other training samples.

One potential solution is to increase the sampling density near the boundary, i.e., $\Delta \vec{x} \to 0$, to avoid such over-fitting. This however increases the complexity of the optimization problem and can cause adverse effects on training. It is also very challenging to locally refine the sampling density for PINNs with CNN-architecture. To resolve this issue without increasing the sample density, we propose BCXN-PINN to impose local structure to govern the near boundary gradient behaviour and restore connectivity between domain boundary and near-boundary samples.

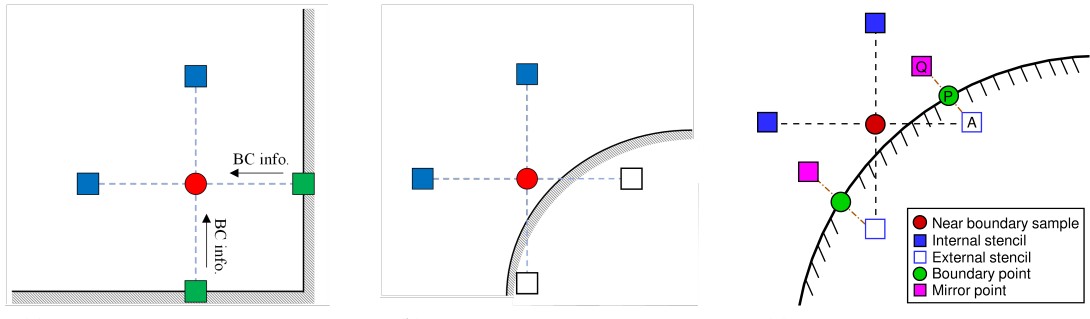

(a) Stencil points connect to boundary     (b) Stencil points fall outside domain     (c) Extrapolation along normal direction

Figure 3: (a)-(b) Schematic of a near-boundary training sample (red circle) and its stencil points (square) typically used for evaluating PDE constraint in *ND-loss*. BC information is passed to stencils in fluid domain (blue square) (a) successfully in regular domain; and (b) unsuccessfully in an irregular domain due to under-defined stencils (unfilled square) outside the domain. (c) Schematic of related definitions under the present BCXN-PINN framework. *BCXN-loss* enforces a constraint across 3 points: external stencil point A, boundary point P, and mirror point Q inside fluid domain.

## 4 METHOD

### 4.1 ENFORCING LINEAR CONSTRAINT AT NEAR-BOUNDARY SAMPLES

This section outlines the proposed BCXN-PINN method. Here we denote the external (out-of-domain) stencil of a near-boundary sample as ES point(s). As per Fig. 3c, the boundary condition $\vec{u}_{BC}$ and field value $\vec{u}_{MI}$ are defined at the boundary point P and a chosen mirror point Q inside the domain, while the field value $\vec{u}_{ES}$ at the ES point A is the value to be determined.

By employing Taylor series expansion on points A and Q relative to point P along $\overline{AQ}$ with respect to local coordinate $n$ (normal direction of solid surface), the following equations can be derived:

$$\vec{u}_{ES} = \vec{u}_{BC} + \overline{AP}\frac{\partial \vec{u}_{BC}}{\partial n} + \frac{\overline{AP}^2}{2}\frac{\partial^2 \vec{u}_{BC}}{\partial n^2} + O(\overline{AP}^3) \tag{3a}$$

$$\vec{u}_{MI} = \vec{u}_{BC} - \overline{PQ}\frac{\partial \vec{u}_{BC}}{\partial n} + \frac{\overline{PQ}^2}{2}\frac{\partial^2 \vec{u}_{BC}}{\partial n^2} + O(\overline{PQ}^3) \tag{3b}$$

where $\overline{AP}$ and $\overline{PQ}$ are the distances between A and P and between P and Q, and $\overline{AQ} = \overline{AP} + \overline{PQ}$. By further manipulation of the above equations, the field value $\vec{u}_{ES}$ at point A can be derived as:

$$\vec{u}_{ES} = \vec{u}_{MI} + (\vec{u}_{BC} - \vec{u}_{MI}) \times \frac{\overline{AQ}}{\overline{PQ}} + O(\overline{AQ}^2) \tag{4}$$

From the above equation, it can be seen that $\vec{u}_{ES}$ is linearly constrained by $\vec{u}_{BC}$ and $\vec{u}_{MI}$. While the mirror point Q can be conveniently chosen at the center of stencil, i.e., the near-boundary sample location, we choose mirror point Q along the normal direction of boundary P (ref. to Fig. 3c) as it is more general and performs better in our experiments. Hence, $\overline{AP} = \overline{PQ}$, and Eq. (4) becomes:

$$\vec{u}_{ES} = 2\vec{u}_{BC} - \vec{u}_{MI} \tag{5}$$

A similar process can be applied for the derivation of corresponding linear constraints for Neumann type and Robin boundary conditions, as detailed in Appendix Section A.1. In addition, PINN methods generally require a routine to sample inside the geometry of interest and at the boundary. As compared to conventional PINNs, BCXN-PINN involves additional steps. We describe the steps to compute the mirror point location (along normal direction) in Appendix Section A.2.

## 4.2 BOUNDARY CONNECTIVITY (BCXN)-LOSS WITH DIRECT FORCING APPROACH

Essentially, the use of a linear constraint can connect under-defined external stencil points to the BCs and the solutions from inside the boundary. We can *directly apply Eq. (5) to compute the field value $\vec{u}_{ES}$ for any external stencil point* during the evaluation of PDE constraint on near-boundary samples with ND-type schemes. This *direct forcing* approach has an association to the direct forcing immersed boundary methods in numerical computing. We can then use the following *df-BCXN-loss*:

$$L_{PINN(df-BCXN)} = L_{PDE(df-BCXN)} \tag{6}$$

where $L_{PDE(df-BCXN)}$ is our BCXN-PINN's PDE loss term whereby the PDE constraint on near-boundary samples is modulated by the direct forcing approach. In such an implementation, the BCs are implicitly "infused" into the training loss through the linear constraint and there is no longer a need to explicitly evaluate the BC loss.

## 4.3 BOUNDARY CONNECTIVITY (BCXN)-LOSS WITH SOFT FORCING APPROACH

The *df-BCXN-loss* strongly enforces the linear constraint during the evaluation of PDE constraint at near-boundary samples. However, the linear condition described by Eq. (5) does not guarantee the best approximation to the local gradient behaviour. The imposition of such an over-simplified constraint can be inappropriate in some scenarios, hence slowing convergence or reducing accuracy in certain instances. To alleviate the issue, we propose to *relax the linear constraint — which may be in conflict with the local gradients — used for propagating BCs to the near-boundary samples in BCXN-PINN by introducing an additional loss term*, in an approach referred to as *soft forcing*.

Let us denote $\vec{u}_{ES_i}$ as the field value computed at the i-th external stencil point by Eq. (5), and $\vec{u}_{ES_i}(\vec{x}; \boldsymbol{w})$ as PINN output at corresponding external stencil point. We then define *BCXN-loss* as:

$$L_{BCXN} = \frac{1}{n_{ES}} \Sigma_{i=1}^{n_{ES}} \left( \vec{u}_{ES_i} - \vec{u}_{ES_i}(\vec{x}; \boldsymbol{w}) \right)^2 \tag{7}$$

for all the ES points $i = 1, \ldots, n_{ES}$ which are required for evaluating the PDE constraint using ND-type schemes. The newly introduced $L_{BCXN}$ specifies the relation at near boundary points, it also explicitly "infuses" the BC information into the PDE samples, hence helping to propagate the correct BC information during training. The new *sf-BCXN-loss* is then defined as:

$$L_{PINN(sf-BCXN)} = \lambda_{PDE} L_{PDE} + \lambda_{BC} L_{BC} + \lambda_{BCXN} L_{BCXN} \tag{8}$$

with additional weight, $\lambda_{BCXN}$, controlling the relative importance of the loss term $L_{BCXN}$ in the loss function. When $\lambda_{BCXN} \to 0$, the $L_{PINN(sf-BCXN)}$ reverts to the conventional *ND*-PINN loss $L_{PINN}$.

## 5 RESULTS

We first study the performance of the proposed BCXN-PINN on diverse 2D and 3D fluid dynamics test cases under forward and inverse problem settings using MLP architecture (inverse problem

results are in Appendix A.12). Then, we demonstrate its efficacy on CNN architecture for both forward and meta-modelling problems. We also demonstrate that BCXN-PINN is broadly applicable to different ND-type training losses, including finite difference-type schemes and coupled-automatic-numerical differentiation (CAN) scheme.

The BCXN-PINN is compared with baseline PINNs which use *ND-loss* and *AD-loss* based on mean squared error (MSE) of the respective PINN solutions' velocity vector $\vec{u}$ relative to ground truth. To facilitate comparison, identical network architectures, initialization distribution, and training settings are employed and summarized in Table 1 and Table 3 in Appendix A.3. For each experiment, we perform and present results based on 10 independent runs with different initialization. The ground truth solutions are obtained by an in-house numerical solver based on the improved divergence-free condition compensated (IDFC) method (Chiu, 2018). The 4 fluid dynamics test cases (schematics in Appendix A.4) presented are:

- **2D semi-circle lid-driven cavity flow,** $Re = 1000$**.** The fluid flow inside the semi-circle cavity is driven by a lid velocity $u_{lid} = 1, v_{lid} = 0$ at the top wall. No-slip condition ($u = v = 0$) is applied to the other wall. There is a primary eddy near the cavity's center.

- **2D lid-driven cavity flow,** $Re = 1000$**.** The fluid flow inside a $1 \times 1$ unit square cavity is driven by a lid velocity $u_{lid} = 1, v_{lid} = 0$ at the top wall. No-slip condition is applied to the other walls. The lid-driven cavity flow has been widely chosen as a benchmark case for many numerical methods due to the complex physics encapsulated within. At $Re = 1000$, there is a primary eddy near the center of the cavity, with secondary and tertiary eddies at the bottom-right and bottom-left regions. This test case has a regular domain, but we chose sample locations such that the near boundary samples have their stencils fall outside the domain to show that the proposed method can be useful for this kind of scenarios.

- **2D wavy channel flow,** $Re = 100$**.** The fluid flow passing through a long wavy channel is studied. The inlet profile at left boundary is defined as $u(0, y) = -\frac{3}{2}y^2 + \frac{3}{2}, v = 0$. A non-slip condition is applied to the top and bottom walls, while outlet boundary conditions ($\frac{\partial u}{\partial x} = \frac{\partial v}{\partial x} = p = 0$) are applied to the right boundary.

- **3D bend tube flow,** $Re = 100$**.** The fluid flow passing through a $90^o$ bending circular tube is studied. The circular channel diameter (D) for the pipe is set as 1, while the mean curvature of the bending curve (R) is set as 3. For this problem, the inlet velocity is specified as a fully developed parabolic profile with bulk velocity equal to unity. The length before the bend curve (L1) is set as 3.5, while the length after the bend curve (L2) is set as 10.

## 5.1 2D AND 3D FORWARD PROBLEMS WITH MLP-ARCHITECTURE BCXN-PINN

In the forward problem experiment, we train PINN models to learn the solution directly from the governing law using different loss functions, i.e., *df-BCXN-* and *sf-BCXN-loss* for BCXN-PINNs, and *AD-* and *ND-loss* for baseline PINNs. We perform 10 independent runs with different initialization for each PINN model. Their learning outcomes on all 4 test cases are shown in Fig. 4, in which we observe a noticeable improvement in solution accuracy ($>2$ orders of magnitude lower $\vec{u}$ MSE) with the BCXN-PINN methods. Fig. 9, Fig. 10 and Fig. 11 in Appendix A.5 visualize the velocity magnitude $\|\vec{u}\|$ and pressure $p$ contours for the median solutions. Visually, the solutions obtained from both *df-BCXN-* and *sf-BCXN-loss* have a very good agreement with the ground truth. The results indicate that our present BCXN-PINN method can successfully learn a good solution with low MSE across a diverse set of problems with irregular domains and complex physics. Both the conventional *ND-* and *AD-loss* PINNs fail to produce a reasonable solution, i.e., they cannot properly learn the correct flow with current sampling density and training iteration. In addition, we apply our method to a i) real-world relevant problem of flow past an airfoil and ii) multi-physics coupled heat and flow problem. Similar improvements are observed, and results are presented in Appendix A.6 and A.7. We discuss the benefits of this connectivity in Appendix A.8.

**Experiment with different sampling density.** We study the 2D lid-driven cavity flow and 3D bend tube channel flow problems to understand the trade-off between the convergence in accuracy and sampling density. We train the same PINN model on a denser (i.e., 3-4x) set of samples until the accuracy of the *ND-* and *AD-loss* PINNs start to improve. Note that denser sampling requires larger training iteration and batch size, and the tuning of these training hyper-parameters can be very time consuming. Fig. 5a-b compare the convergence trends in terms of $\vec{u}$ MSE between sparse

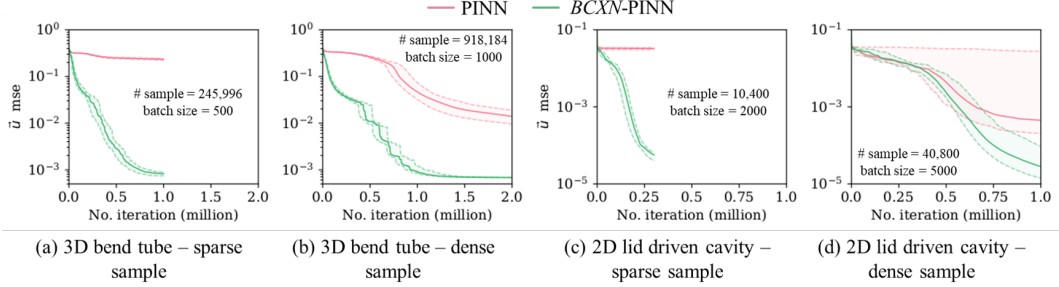

Figure 4: Distribution of $\vec{u}$ MSE from BCXN-PINNs (*df-BCXN-* and *sf-BCXN-loss*) and PINNs (*ND-* and *AD-loss*) while using MLP-architecture for the 4 forward problem test cases.

Figure 5: PINNs learning the solution of (a-b) 3D bend tube channel flow and (c-d) 2D lid-driven cavity flow under sparse and dense training scenarios respectively. The convergence trends of $\vec{u}$ MSE are plotted. Bold lines indicate the median convergence path, and the shaded areas indicate the inter-percentile range from their $10^{th} - 90^{th}$ percentiles.

(i.e., the one shown in previous section) and dense sampling training scenarios for the 3D bend tube channel flow problem. BCXN-PINN learns more accurate solutions (>1 order of magnitude lower $\vec{u}$ MSE) than the conventional PINNs with faster speed (50% less training iterations), while using fewer (<50%) training samples. A similar trend is observed for the 2D lid-driven cavity flow problem (Fig. 5c), where we observe that BCXN-PINN quickly learns an accurate solution under sparse sample training. Although all PINN models can eventually achieve improved accuracy in this test case, Fig. 5d indicates that it is much harder and slower to learn from a dense set of samples, thereby demonstrating the benefits of BCXN-PINN. Additional details are in Appendix A.9.

**Hyper-parameter tuning.** We use settings as per Table 1 in Appendix A.3, and do not exhaustively search for best set of hyper-parameters for the individual PINN models. Apart from sampling density, we observe that PINN learning is sensitive to NN capacity, batch size and learning rate. Another important hyper-parameter for PINNs is the relative weights $\lambda_{PDE}$ and $\lambda_{BC}$ in the loss function $L_{PINN}$. Specifically, we can further decompose $\lambda_{PDE}$ into $\lambda_{DIV}$ for the divergence free condition (Eq. (1a)) and $\lambda_{MOM}$ for the conservation of momentum equations (Eq. (1b)) in the PDE loss. A properly tuned $\lambda_{DIV}$ can further improve the solution accuracy of BCXN-PINN by another half order of magnitude in $\vec{u}$ MSE for the 2D wavy channel flow ($\lambda_{DIV} = 5$) and 3D bend tube channel flow ($\lambda_{DIV} = 20$). We provide more details in Appendix A.11.

## 5.2 CNN-ARCHITECTURE BCXN-PINN: FORWARD AND META-MODELLING PROBLEMS

The BCXN-PINN method does not pose any requirement on the differentiable property of the networks hence it can be freely implemented in any neural network architecture, including CNN. We demonstrate this by using the *df-BCXN-loss* for learning the solution of a 2D semi-circle lid-driven cavity flow problem with U-Net architecture. We show that the *df-BCXN-loss* can benefit CNN-architecture PINNs by remedying training issues stemming from being unable to exactly resolve complex geometries due to the use of a discretized input. We also test the BCXN-PINN method with *df-BCXN-loss* on a 2D lid-driven cavity flow problem to demonstrate its superior performance. The U-Net architecture and training settings are summarized in Appendix A.3 Table 3. We train the forward BCXN-PINN models on a single discretized 2D input which consists of 3 channels, namely the x-, y-coordinate, and an indicator to differentiate fluid and non-fluid domain.

The performance of BCXN-PINN and baseline PINN (i.e., using *ND-loss*) on the 2 test cases are compared in Fig. 6. Like the MLP experiment, BCXN-PINN has a noticeable improvement in the solution accuracy (2-3 orders of magnitude lower $\vec{u}$ MSE). Visually, the solutions obtained from the

BCXN-PINN have a very good agreement with the ground truth, while the baseline PINN fails to produce a reasonable solution (in Appendix A.13 Fig. 19). Additional results for a transient version of 2D semi-circle lid-driven cavity flow problem are presented in Appendix A.15 along with details on the impact of Re on model error and training time in Appendix A.16.

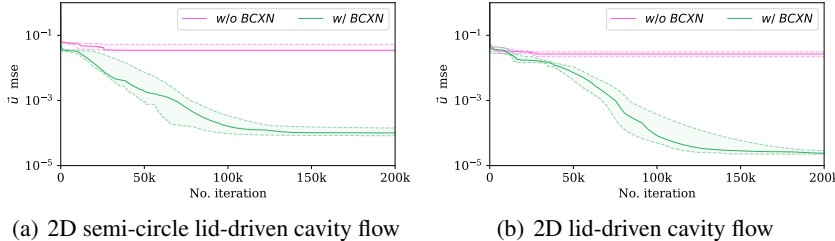

(a) 2D semi-circle lid-driven cavity flow          (b) 2D lid-driven cavity flow

Figure 6: The convergence trends of $\vec{u}$ MSE for PINN models using CNN architecture are plotted.

**Meta-PINN model.** The CNN-architecture BCXN-PINN represent the entire spatial domain as a single input. We can train the model on multiple samples, each representing a different scenario such as changes in geometry or physical behaviour. Such a meta-PINN model can be used to predict new scenarios. Hence, we further study the ability of BCXN-PINN to learn the solutions to multiple 2D semi-circle lid-driven cavity flow scenarios concurrently (spanning $Re = 50 - 1000$). The model architecture and training setting is given in Appendix A.3 Table 3. The *df-BCXN-PINN* model is used to predict $Re = 300, 500, 600, 700, 900$ scenarios which are not used during training. The results are summarized in Fig. 7. The present BCXN-PINN method successfully learns a good solution with low $\vec{u}$ MSE (i.e., <1e-4) for most of the $Re$ scenarios, and achieves good prediction accuracy.

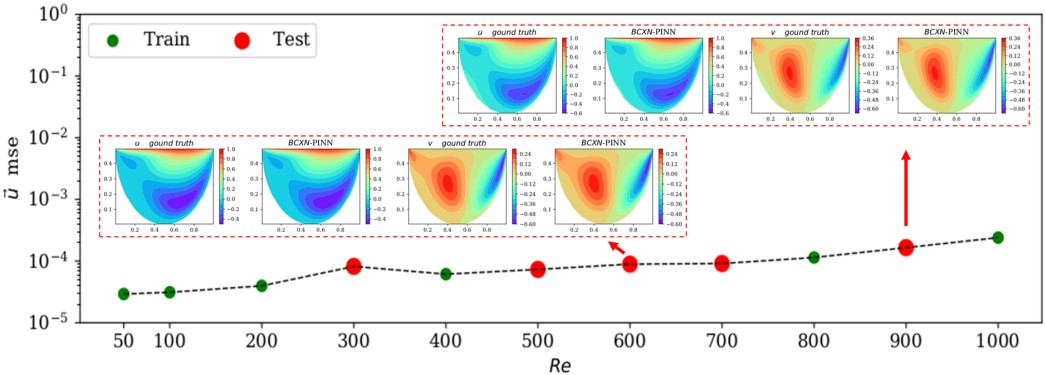

Figure 7: BCXN-PINN learning the solutions of multiple 2D semi-circle lid-driven cavity flow scenarios concurrently, using CNN architecture. The $\vec{u}$ MSE between ground truth and BCXN-PINN solution for train and test $Re$ are plotted. Inset are *u*- and *v*-velocity contours of the ground truth and BCXN-PINN solutions for $Re = 600$ and $Re = 900$.

## 6    CONCLUSION

We present a BCXN-PINN method which can more efficiently learn the solution to PDEs for complex geometry with fewer training samples. This is accomplished by enforcing a linear constraint during the training implicitly and explicitly via direct forcing and soft forcing approaches. While linear constraints are applied in this work due to truncation of the Taylor series expansion in the derivation, it is worth studying if different, higher order approximations can also be utilized to achieve better performance in future work. Nonetheless, our comprehensive experimental studies demonstrate practical advantages of BCXN-PINN on diverse test problems in fluid dynamics, spanning both forward and inverse problems in 2D and 3D, and a meta-model problem, improving the accuracy of solutions by orders of magnitude even while requiring much less training iterations and samples.

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

# A APPENDIX

## A.1 DERIVATION OF LINEAR CONSTRAINT FOR NEUMANN TYPE AND ROBIN TYPE BOUNDARY CONDITION

Based on Eq. 3, the linear constraint for Neumann type boundary condition can be similarly derived utilizing Eqn 3(a) - 3(b):

$$\vec{u}_{ES} - \vec{u}_{MI} = \overline{AQ}\frac{\partial \vec{u}_{BC}}{\partial n} + \frac{(\overline{AP}^2 - \overline{PQ}^2)}{2}\frac{\partial^2 \vec{u}_{BC}}{\partial n^2} + O(\overline{AQ}^3) \tag{9}$$

or

$$\vec{u}_{ES} = \vec{u}_{MI} + \overline{AQ}\frac{\partial \vec{u}_{BC}}{\partial n} + O(\overline{AP}^2 - \overline{PQ}^2) \tag{10}$$

For the homogeneous Neumann boundary condition ($\frac{\partial \vec{u}_{BC}}{\partial n} = 0$), Eq. (10) can be further simplified as

$$\vec{u}_{ES} = \vec{u}_{MI} \tag{11}$$

A similar treatment can be used to derive a corresponding linear constraint for problems with Robin type boundary condition:

$$\vec{u}_{BC} + \beta \frac{\partial \vec{u}_{BC}}{\partial n} = c \tag{12}$$

It is then possible to obtain an appropriate expression by substituting the above expression into Eq. (4):

$$\vec{u}_{ES} = \vec{u}_{MI} + ((c - \beta \frac{\partial \vec{u}_{BC}}{\partial n}) - \vec{u}_{MI}) \times \frac{\overline{AQ}}{\overline{PQ}} \tag{13}$$

In the above, $\frac{\partial \vec{u}_{BC}}{\partial n}$ is approximated by Eq. (10).

We forward demonstrate the ability of the current proposed method to handle various combinations of Dirichlet and Neumann type boundary conditions in Appendix A.14.

## A.2 PROCEDURE TO EVALUATE THE FIELD VALUE $\vec{u}_{MI}$ AT MIRROR POINT

Note that PINN methods in general require a routine to sample from inside the geometry of interest as well as the boundary. As compared to conventional PINNs, BCXN-PINN involves additional steps. We first describe the steps to compute the mirror point location (along normal direction):

- **Determine whether a stencil is external.** In this study, we first construct a level set function $\phi$ of the geometry of interest (Osher & Paragios, 2003), where the shortest distance $\overline{AP}$ between the stencil point A and boundary point P can be found. From the sign of level set function, we can determine whether a stencil is external.

- **Compute mirror point location for external stencil.** Once $\overline{AP}$ is determined, we compute the location of mirror point Q inside the fluid domain, with the distance $\overline{AP} = \overline{PQ}$.

In practice, given a fixed set of training samples, all the external stencil points and their mirror locations can be pre-computed. On the other hand, the field value $\vec{u}_{MI}$ at mirror points depends on the PINN's output and are evaluated inside a training iteration:

- **MLP-architecture PINN.** The $\vec{u}_{MI}$ value can be directly obtained by evaluating $\vec{u}_{MI}(\vec{x}; \boldsymbol{w})$ at the mirror point.

- **CNN-architecture PINN.** The location of mirror point may not coincide with the CNN grid. In such case, the following inverse-distance-weighted interpolation function has been utilized to obtain the $\vec{u}_{MI}$ value at mirror point (Chiu & Poh, 2021).

$$u(\tilde{x}, \tilde{y}) = \Sigma u(x, y) \phi' \left( \frac{x - \tilde{x}}{h} \right) \phi' \left( \frac{y - \tilde{y}}{h} \right) \tag{14a}$$

$$\phi' \left( \frac{x - \tilde{x}}{h} \right) = \phi \left( \frac{x - \tilde{x}}{h} \right) / \Sigma \phi \left( \frac{x - \tilde{x}}{h} \right) \tag{14b}$$

$$\phi' \left( \frac{y - \tilde{y}}{h} \right) = \phi \left( \frac{y - \tilde{y}}{h} \right) / \Sigma \phi \left( \frac{y - \tilde{y}}{h} \right) \tag{14c}$$

$$\phi(r) = \begin{cases} \frac{1}{8} \left( 3 - 2|r| + \sqrt{1 + 4|r| + 4r^2} \right), & |r| \leq 1 \\ \frac{1}{8} \left( 5 - 2|r| - \sqrt{-7 + 12|r| - 4r^2} \right), & 1 < |r| \leq 2 \\ 0, & |r| > 2 \end{cases} \tag{14d}$$

$(\tilde{x}, \tilde{y})$ is the location of mirror point to be interpolated, while $(x, y)$ is the location of training samples inside the fluid domain. In this study, $h$ is chosen as $\Delta x$, which leads to a $2\Delta x$ interpolation radius. Once $\vec{u}_{MI}$ and $\vec{u}_{BC}$ are evaluated, we can compute $\vec{u}_{ES}$ based on Eq. (5) during training.

A.3 MLP AND CNN EXPERIMENTAL SETTINGS

Table 1: PINN (MLP) architecture and training settings used for experiments in Section 5.1

| Forward Problem | 1 | 2 | 3 | 4 |
|---|---|---|---|---|
| Test case | 2D semi-circle lid-driven cavity flow, $Re = 1000$ | 2D lid-driven cavity flow, $Re = 1000$ | 2D wavy channel flow, $Re = 100$ | 3D bend tube channel flow, $Re = 100$ |
| *Governing physics eqns.* | 2D *incompressible NS eqns.* | 2D *incompressible NS eqns.* | 2D *incompressible NS eqns.* | 3D *incompressible NS eqns.* |
| MLP architecture *for BCXN-PINN, ND-PINN & AD-PINN* | $(x,y)-$ 64–30–30–30– $[30–30–30–(\hat{u})$, 30–30–30–$(\hat{v})$, 30–30–30–$(\hat{p})]$ | $(x,y)-$ 128–30–30–30– $[30–30–30–(\hat{u})$, 30–30–30–$(\hat{v})$, 30–30–30–$(\hat{p})]$ | $(x,y)-$ 128–40–40–40– $[40–40–40–(\hat{u})$, 40–40–40–$(\hat{v})$, 40–40–40–$(\hat{p})]$ | $(x,y,z)-$ 128–30–30–30– $[30–30–30–(\hat{u})$, 30–30–30–$(\hat{v})$, 30–30–30–$(\hat{w})$, 30–30–30–$(\hat{p})]$ |
| ND scheme *for BCXN-PINNs & ND-PINN* | CAN & finite difference | CAN | CAN | Finite difference |
| Training sample *(equidistantly spaced samples @ fluid domain & boundary)* | $3,930 + 300$ | $10,000 + 400$ | $14,400 + 800$ | $223,612 + 22,384$ |
| Batch size | $1,000$ | $2,000$ | $1,000$ | $500$ |
| Training iterations | $300,000$ | $300,000$ | $300,000$ | $1,000,000$ |

(I) For the MLP architecture, the numbers in between input and output represent the number of nodes in each hidden layer. For example, $(x)$–64–20–20–20–$(\hat{u})$ indicates a single input $x$, followed by 4 hidden layers with 64, 20, 20 and 20 nodes in each layer, and a single output $\hat{u}$.

(II) We incorporate the sinusoidal mapping (Wong et al., 2022) into the first hidden layer of PINN and initialize its weights by sampling from a normal distribution $N(0, \sigma^2), \sigma = 1$. The subsequent hidden layers use "sine" activation, except a "linear" activation function is used in the final (output) layer, and their weights are initialized by He uniform distribution.

(III) Batch size: number of random sample used for 1 evaluation of $L_{PINN}(= \lambda_{PDE} L_{PDE} + \lambda_{BC} L_{BC})$, $L_{PINN(df-BCXN)}(= \lambda_{PDE} L_{PDE(df-BCXN)})$, and $L_{PINN(sf-BCXN)}(= \lambda_{PDE} L_{PDE} + \lambda_{BCXN} L_{BCXN} + \lambda_{BC} L_{BC})$. We used a default $\lambda_{PDE} = 1$, $\lambda_{BC} = 1$, and $\lambda_{BCXN} = 1$, unless otherwise mentioned.

(IV) A training iteration: 1 evaluation of $L_{PINN}$, $L_{PINN(df-BCXN)}$ or $L_{PINN}(sf - BCXN)$ for back-propagating the weight gradients. We use an initial learning of 1e-3 and reduce it on plateauing, until a min. learning rate of 5e-6 is reached.

Table 2: PINN (MLP) architecture and training settings used for experiments described in Appendix

| Forward Problem | 5 | 6 | 7 |
|---|---|---|---|
| Test case | 2D flow past airfoil shape, $Re = 500$ | 2D lid-driven cavity flow and heat transfer with a cylinder inside, $Re = 100$ | 1D convection diffusion problem |
| *Governing physics eqns.* | *2D incompressible NS eqns.* | *2D incompressible NS eqns.* + energy equation for temperature | $20u_x = u_{xx}$ |
| MLP architecture *for BCXN-PINN, ND-PINN & AD-PINN* | $(x,y)-$ $64-40-40-40-$ $[40-40-40-(\hat{u}),$ $40-40-40-(\hat{v}),$ $40-40-40-(\hat{p})]$ | $(x,y)-$ $64-30-30-30-$ $[30-30-30-(\hat{u}),$ $30-30-30-(\hat{v}),$ $30-30-30-(\hat{p}),$ $30-30-30-(\hat{\Theta})]$ | $(x,y)-$ $64-40-40-40-(\hat{u})$ |
| ND scheme *for BCXN-PINNs & ND-PINN* | CAN | CAN | CAN & finite difference |
| Training sample *(equidistantly spaced samples @ fluid domain & boundary)* | $78,690 + 1,450$ | $2,184 + 250$ | $30 + 2$ |
| Batch size | 500 | 1,000 | 15 |
| Training iterations | 500,000 | 100,000 | 50,000 |

(I) For the MLP architecture, the numbers in between input and output represent the number of nodes in each hidden layer. For example, $(x)-64-20-20-20-(\hat{u})$ indicates a single input $x$, followed by 4 hidden layers with 64, 20, 20 and 20 nodes in each layer, and a single output $\hat{u}$.

(II) We incorporate the sinusoidal mapping (Wong et al., 2022) into the first hidden layer of PINN and initialize its weights by sampling from a normal distribution $N(0, \sigma^2), \sigma = 1$. The subsequent hidden layers use "sine" activation, except a "linear" activation function is used in the final (output) layer, and their weights are initialized by He uniform distribution.

(III) Batch size: number of random sample used for 1 evaluation of $L_{PINN}(= \lambda_{PDE} \ L_{PDE} + \lambda_{BC} \ L_{BC})$, $L_{PINN(df-BCXN)}(= \lambda_{PDE} \ L_{PDE(df-BCXN)})$, and $L_{PINN(sf-BCXN)}(= \lambda_{PDE} \ L_{PDE} + \lambda_{BCXN} \ L_{BCXN} + \lambda_{BC} \ L_{BC})$. We used a default $\lambda_{PDE} = 1$, $\lambda_{BC} = 1$, and $\lambda_{BCXN} = 1$, unless otherwise mentioned.

(IV) A training iteration: 1 evaluation of $L_{PINN}$, $L_{PINN(df-BCXN)}$ or $L_{PINN}(sf-BCXN)$ for back-propagating the weight gradients. We use an initial learning of 1e-3 and reduce it on plateau-ing, until a min. learning rate of 5e-6 is reached.

Table 3: PINN (CNN) architecture and training settings used in the experiments in Section 5.2

| Problem type | Forward | Forward | Meta-modelling |
|---|---|---|---|
| Test case | 2D semi-circle lid-driven cavity flow, $Re = 1000$ | 2D lid-driven cavity flow, $Re = 1000$ | 2D semi-circle lid-driven cavity flow, $Re = 1000$ |
| *Governing physics eqns.* | 2D *incompressible NS eqns.* | 2D *incompressible NS eqns.* | 2D *incompressible NS eqns.* |
| CNN (U-Net) architecture *for BCXN-PINN & ND-PINN* | $(\boldsymbol{X}_{(56\times112\times3)})$–, 8–8–↓–16–16–↓–, 32–32–↑–64–64–↑ –, 32–32–↑ –16–16–↑–[, 8–8–8–$(\hat{\boldsymbol{U}}_{u,56\times112})$, 8–8–8–$(\hat{\boldsymbol{U}}_{v,56\times112})$, 8–8–8–$(\hat{\boldsymbol{U}}_{p,56\times112})$ | $(\boldsymbol{X}_{(104\times104\times3)})$–, 8–8–↓–16–16–↓–, 32–32–↑–64–64–↑ –, 32–32–↑ –16–16–↑–[, 8–8–8–$(\hat{\boldsymbol{U}}_{u,104\times104})$, 8–8–8–$(\hat{\boldsymbol{U}}_{v,104\times104})$, 8–8–8–$(\hat{\boldsymbol{U}}_{p,104\times104})$ | $(\boldsymbol{X}_{(56\times112\times4)})$–, 8–8–↓–16–16–↓–, 32–32–↑–64–64–↑ –, 32–32–↑ –16–16–↑–[, 8–8–8–$(\hat{\boldsymbol{U}}_{u,56\times112})$, 8–8–8–$(\hat{\boldsymbol{U}}_{v,56\times112})$, 8–8–8–$(\hat{\boldsymbol{U}}_{p,56\times112})$ |
| Kernel size | $4 \times 4$ | $6 \times 6$ | $6 \times 6$ |
| ND scheme *for BCXN-PINNs & ND-PINN* | Finite difference | Finite difference | Finite difference |
| Training sample */ batch size* | 1 / 1 | 1 / 1 | 6 / 1 |
| Training iterations | $200,000$ | $200,000$ | $500,000$ |

(I) For the U-net architecture, the numbers in between input and output represent the number of filters in each hidden layer. Each hidden layer consists of convolution operation, batch normalization, and nonlinear "sine" activation. Also, ↓ represents a $2 \times 2$ max-pooling operation, and ↑ represents a $2 \times 2$ up-sampling operation. The network weights are initialized by He uniform distribution.

(II) Batch size: number of sample used for 1 evaluation of $L_{PINN}(= \lambda_{PDE} L_{PDE} + \lambda_{BC} L_{BC})$ and $L_{PINN(df-BCXN)}(= \lambda_{PDE} L_{PDE(df-BCXN)})$. We used a default $\lambda_{PDE} = 1$ and $\lambda_{BC} = 1$, unless otherwise mentioned.

(III) A training iteration: 1 evaluation of $L_{PINN}$, $L_{PINN(df-BCXN)}$ or $L_{PINN(sf-BCXN)}$ for backpropagating the weight gradients. We use an initial learning of 1e-3 and reduce it on plateauing, until a min. learning rate of 5e-6 is reached.

(IV) The semi-circle and regular lid-driven cavity geometries consist of 3,930 and 10,000 interior points which belong to the fluid domain respectively.

Table 4: PINN (CNN) architecture and training settings used in the experiments described in Appendix

| Problem type | Forward | Forward |
|---|---|---|
| Test case | 2D Taylor-Couette flow, $Re = 1000$ | 2D transient semi-circular lid-driven cavity flow, $Re = 500$ |
| *Governing physics eqns.* | 2D *incompressible NS eqns.* | 2D *transient incompressible NS eqns.* |
| CNN (U-Net) architecture *for BCXN-PINN & ND-PINN* | $(\boldsymbol{X}_{(56\times56\times3)})$–, 8–8–↓–16–16–↓–, 32–32–↑–64–64–↑ –, 32–32–↑ –16–16–↑–[, 8–8–8–$(\hat{\boldsymbol{U}}_{u,56\times56})$, 8–8–8–$(\hat{\boldsymbol{U}}_{v,56\times56})$, 8–8–8–$(\hat{\boldsymbol{U}}_{p,56\times56})$ | $(\boldsymbol{X}_{(56\times112\times4)})$–, 8–8–↓–16–16–↓–, 32–32–↑–64–64–↑ –, 32–32–↑ –16–16–↑–[, 8–8–8–$(\hat{\boldsymbol{U}}_{u,56\times112})$, 8–8–8–$(\hat{\boldsymbol{U}}_{v,56\times112})$, 8–8–8–$(\hat{\boldsymbol{U}}_{p,56\times112})$ |
| Kernel size | $4 \times 4$ | $4 \times 4$ |
| ND scheme *for BCXN-PINNs & ND-PINN* | Finite difference | Finite difference |
| Training sample */ batch size* | 1 / 1 | 21 / 2 |
| Training iterations | $50,000$ | $100,000$ |

(I) For the U-net architecture, the numbers in between input and output represent the number of filters in each hidden layer. Each hidden layer consists of convolution operation, batch normalization, and nonlinear "sine" activation. Also, ↓ represents a $2 \times 2$ max-pooling operation, and ↑ represents a $2 \times 2$ up-sampling operation. The network weights are initialized by He uniform distribution.

(II) Batch size: number of sample used for 1 evaluation of $L_{PINN}(= \lambda_{PDE} \ L_{PDE} + \lambda_{BC} \ L_{BC})$ and $L_{PINN(df-BCXN)}(= \lambda_{PDE} \ L_{PDE(df-BCXN)})$. We used a default $\lambda_{PDE} = 1$ and $\lambda_{BC} = 1$, unless otherwise mentioned.

(III) A training iteration: 1 evaluation of $L_{PINN}$, $L_{PINN(df-BCXN)}$ or $L_{PINN(sf-BCXN)}$ for backpropagating the weight gradients. We use an initial learning of 1e-3 and reduce it on plateauing, until a min. learning rate of 5e-6 is reached.

(IV) The Taylor-Couette geometry consists of 1,216 interior points in the fluid domain while the transient semi-circle lid-driven cavity problem consists of 3,930 interior spatial points and 21 temporal snapshots ($\Delta t = 0.1$) respectively.

In addition, we summarize the training results across different problems and model architectures in 2 Summary Tables in Appendix A.18.

A.4 PROBLEM SCHEMATIC FOR FLUID DYNAMIC CASE STUDIES

The following figure is schematics of the fluid dynamic case studies in this work, depict the 2D semi-circle lid-driven cavity, 2D lid-driven cavity, 2D wavy channel and 3D bend tube respectively.

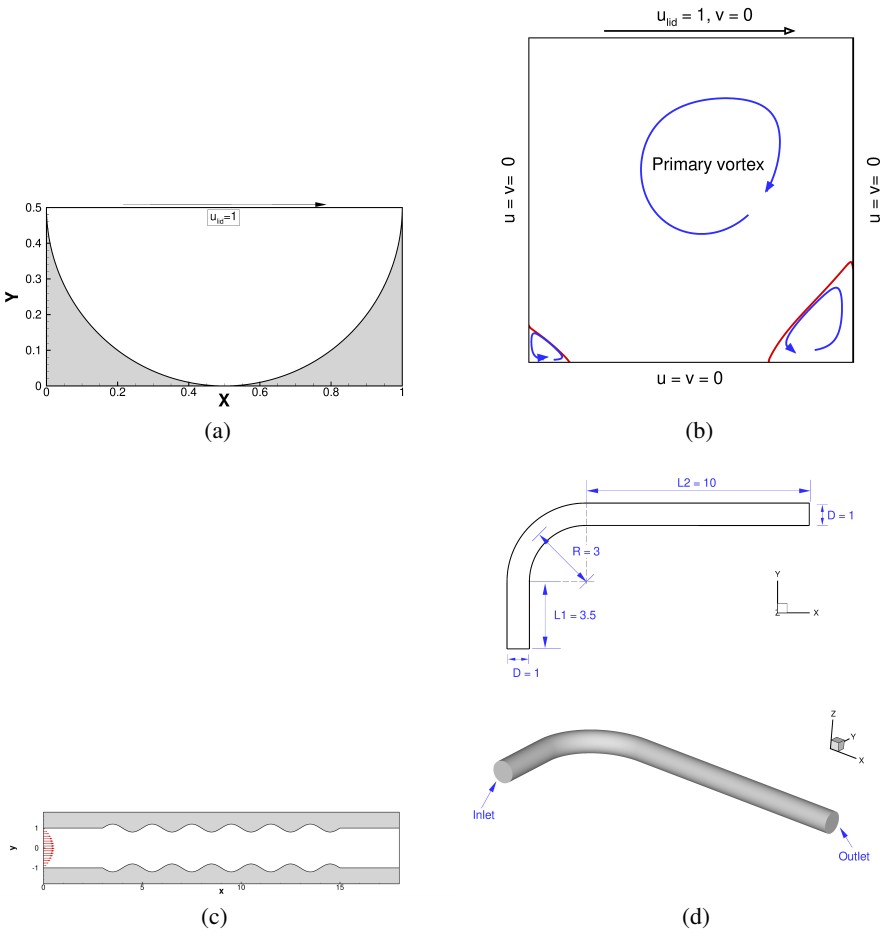

Figure 8: Schematic of the investigated problem. (a) 2D semi-circle lid-driven cavity flow problem; (b) 2D lid-driven cavity flow problem; (c) 2D wavy channel flow problem; (d) 3D bend tube flow problem.

A.5   CONTOUR PLOTS FOR FORWARD PROBLEM WITH MLP-ARCHITECTURE BCXN-PINN

The following figures are visualizations of the velocity and pressure contours obtained from the problems described in Section 5.1. Fig. 9, Fig. 10 and Fig. 11 are visualizations of the velocity magnitude $\|\vec{u}\|$ and pressure $p$ contours for the median solutions.

A.6   ADDITIONAL FORWARD PROBLEM ON 2D FLOW PAST AIRFOIL SHAPE, $Re = 500$

To further validate the applicability and efficacy of the present *BCXN-loss* method for more real-world complex geometries, we also applied the MLP-architecture BCXN-PINN to the modelling of incompressible flow past an airfoil, similar to other previous work (Pfaff et al., 2020).

As shown in Fig. 12, an airfoil with a NACA0012 profile is placed in a modelling domain $(x, y) = (-3 \sim 5, -2 \sim 2)$ with zero angle of attack. A uniform inlet velocity, $u_{inlet} = 1$, is specified at $x = -3$, and a zero pressure outlet boundary condition is specified at $x = 5$. Typical slip boundary conditions are specified at the side walls. The PINN model details are described in Table 3. Briefly, the total training samples for this problem is 79,886, with the sample spacing in the domain corresponding to $\Delta x = \Delta y = 1/50$. In addition, the weight for *BCXN-loss* is set to be 0.1. Contour plots of the obtained velocities for these experiments are shown in Fig. 13.

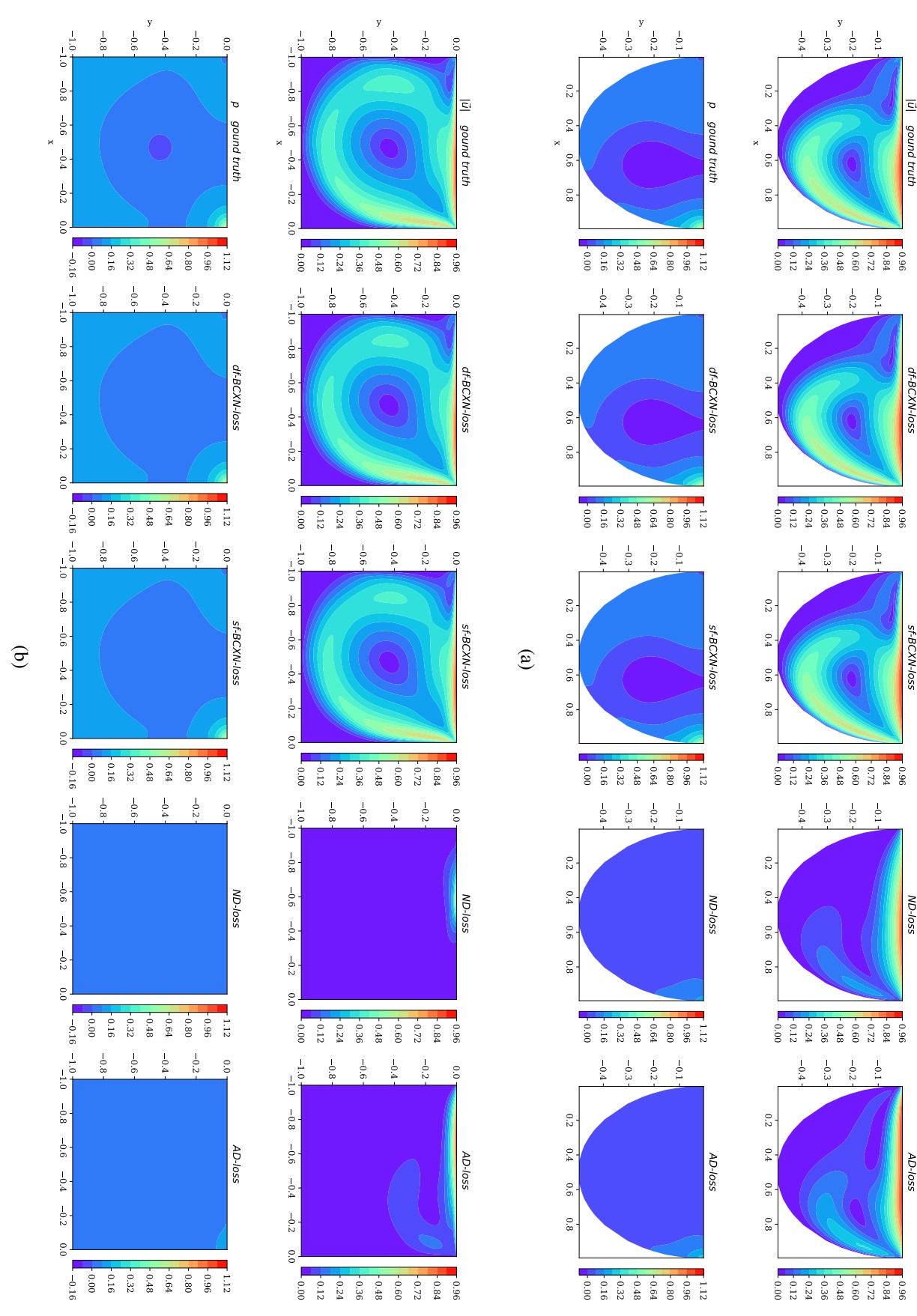

Figure 9: Comparison of velocity magnitude $\|\vec{u}\|$ and pressure $p$ contour between the ground truth and PINN solutions (median MSE from 10 runs), for the (a) 2D semi-circle lid-driven cavity flow problem, $Re = 1000$, and (b) 2D lid-driven cavity flow problem, $Re = 1000$.

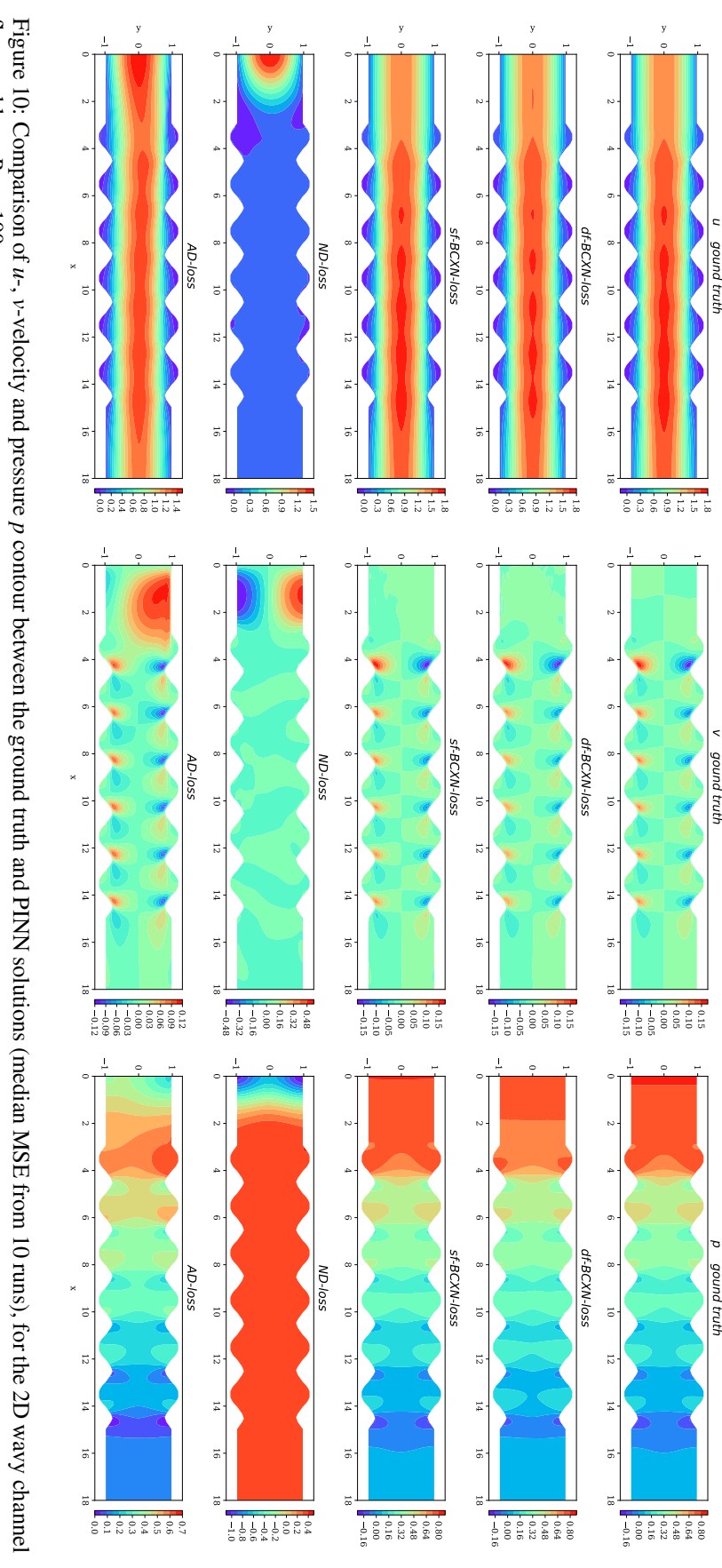

Figure 10: Comparison of $u$-, $v$-velocity and pressure $p$ contour between the ground truth and PINN solutions (median MSE from 10 runs), for the 2D wavy channel flow problem, $Re = 100$.

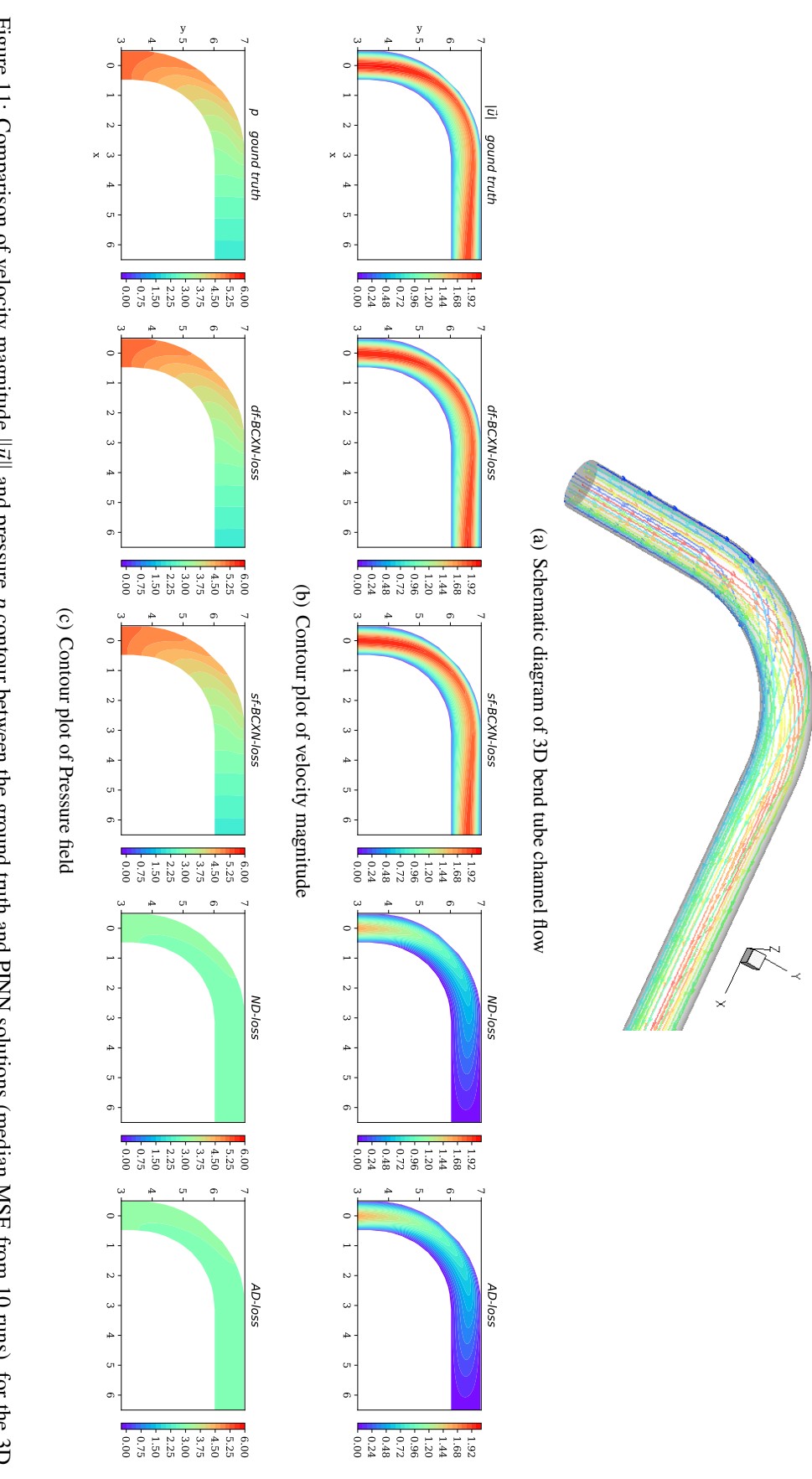

Figure 11: Comparison of velocity magnitude $\|\vec{u}\|$ and pressure $p$ contour between the ground truth and PINN solutions (median MSE from 10 runs), for the 3D bend tube channel flow problem, $Re = 100$. Plots are of a 2D cross-section at $z = 0.0125$.

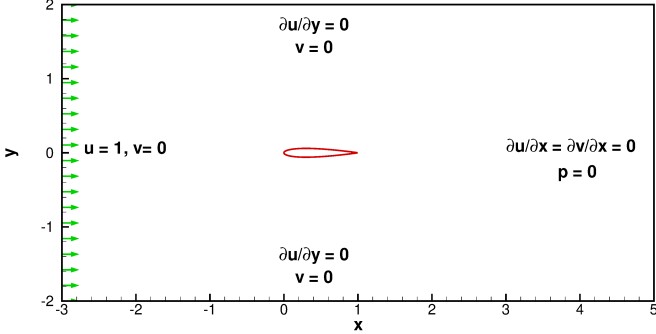

Figure 12: Schematic of 2D flow past airfoil shape problem.

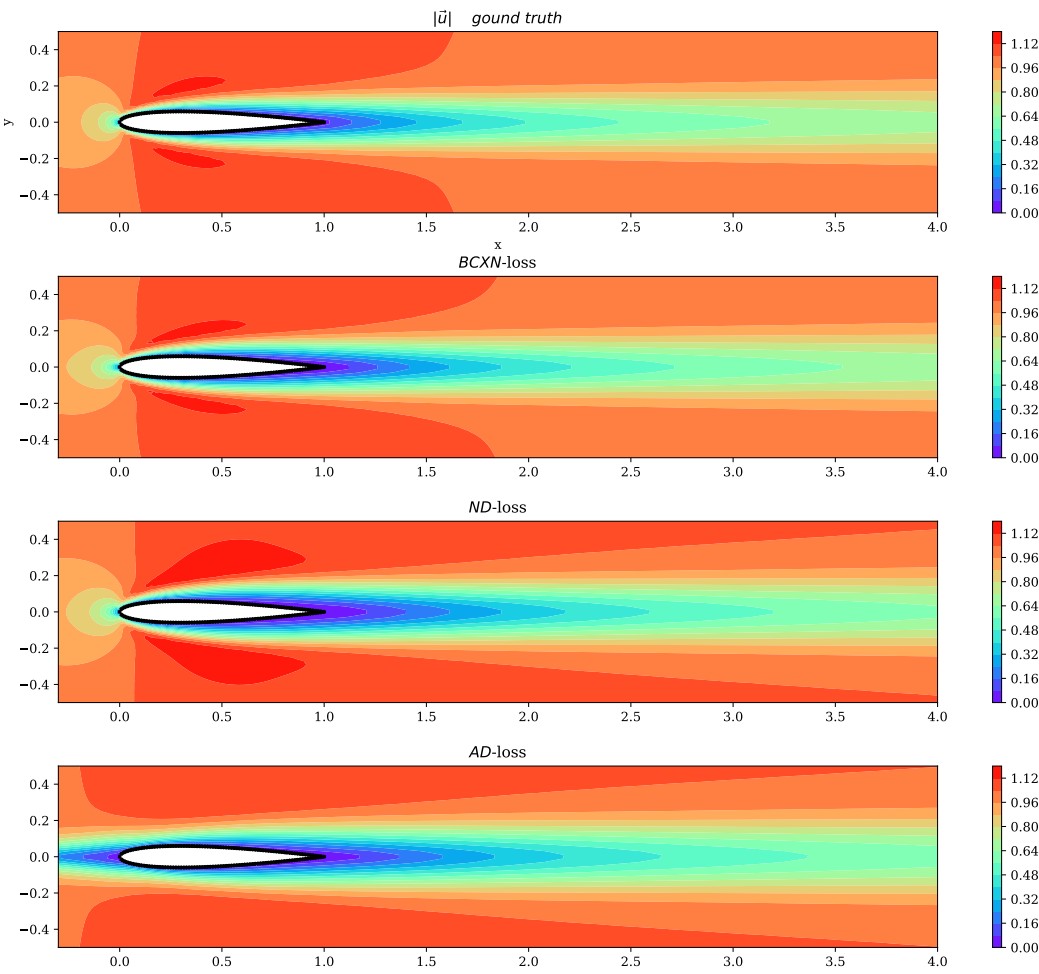

Figure 13: Contour plots of velocity fields obtained from 4 models: (Top) CFD, (2nd from Top) *BCXN-loss*, (2nd from Bottom) *ND-loss*, (Bottom) *AD-loss*.

Based on these numerical experiments, we similarly notice that the incorporation of a *BCXN-loss* improves the PINN's ability to learn the flow profile around the airfoil by an order of magnitude. Quantitative results are provided in Table 5.

Table 5: Model Performance for NACA 0012

| Model | MSE | Rel. Error |
|---|---|---|
| *sf-BCXN-CAN* | $4.65 \times 10^{-5}$ | $8.10 \times 10^{-2}$ |
| *CAN* | $3.32 \times 10^{-4}$ | $1.92 \times 10^{-1}$ |
| *sf-BCXN-ND* | $5.03 \times 10^{-5}$ | $6.93 \times 10^{-2}$ |
| *ND* | $3.54 \times 10^{-4}$ | $1.84 \times 10^{-1}$ |
| *AD* | $1.18 \times 10^{-3}$ | $3.17 \times 10^{-1}$ |

### A.7 ADDITIONAL FORWARD PROBLEM ON 2D LID-DRIVEN CAVITY FLOW WITH HEAT TRANSFER, $Re = 100$

We further proposed to validate the applicability of the proposed method to a multi-physics example. In this instance, we applied the MLP-architecture BCXN-PINN to the modelling of incompressible flow with heat transfer under Boussinesq approximation:

$$\nabla \cdot \vec{u} = 0 \tag{15a}$$

$$(\vec{u} \cdot \nabla) \vec{u} = Re^{-1} \Delta \vec{u} - \nabla p + Ri \Theta \vec{e} \tag{15b}$$

$$(\vec{u} \cdot \nabla) \Theta = Re^{-1} Pr^{-1} \Delta \Theta \tag{15c}$$

in the above, $Pr$ is Prandtl number, $Ri$ is Richardson number, and $\vec{e} = (0, 1)$.

In the present study, there is a 2D lid-driven cavity domain with a stationary cylinder of radius 0.2 in the centre of the domain, while $Pr$ is set as 0.7 and $Ri$ is set as 1. For the boundary conditions, the bottom wall is set as the hot surface($\Theta = 1$) and top wall is set as the cold surface ($\Theta = 0$). In addition, the side walls and cylinder surface are specified as adiabatic boundary conditions. The PINN model details are described in detail in Table 3 while final results are presented in Table 6.

Table 6: Model Performance

| MLP-Model | Variable | $d\Theta/dn = 0$ | | $d\Theta/dn = 1$ | |
|---|---|---|---|---|---|
| | | MSE | Rel Err | MSE | Rel Err |
| *df-BCXN-CAN* | $\vec{u}$ | $1.31 \times 10^{-4}$ | $3.42 \times 10^{-4}$ | $4.45 \times 10^{-4}$ | $2.00 \times 10^{-2}$ |
| | $\vec{\Theta}$ | $5.06 \times 10^{-2}$ | $3.77 \times 10^{-2}$ | $8.81 \times 10^{-2}$ | $2.64 \times 10^{-1}$ |
| *CAN* | $\vec{u}$ | $2.32 \times 10^{-4}$ | $4.40 \times 10^{-4}$ | $6.87 \times 10^{-4}$ | $2.02 \times 10^{-2}$ |
| | $\vec{\Theta}$ | $6.76 \times 10^{-2}$ | $4.27 \times 10^{-2}$ | $1.07 \times 10^{-1}$ | $2.66 \times 10^{-1}$ |
| *AD* | $\vec{u}$ | $3.41 \times 10^{-2}$ | $1.85 \times 10^{-1}$ | $3.89 \times 10^{-2}$ | $2.21 \times 10^{-1}$ |
| | $\vec{\Theta}$ | $8.23 \times 10^{-1}$ | $8.77 \times 10^{-1}$ | $8.65 \times 10^{-1}$ | $8.80 \times 10^{-1}$ |

Based on our numerical experiments, the *BCXN-loss* appears to also yield some benefit on multi-physics problems, with consistently lower MSE and relative L2 errors than the baseline comparisons. However, more thorough experiments need to be conducted to fully validate situations whereby the *BCXN-loss* may be more beneficial.

### A.8 CASE STUDY OF INFORMATION PROPAGATION

Information propagation from the boundary to the inner domain has been suggested to be critical for successful PINN modelling. Hence, we investigate the evolution of the PDE residual across multiple training runs for the 2D semi-circle lid-driven cavity with and without the imposition of the *BCXN-loss* to better understand the impact of enhanced connectivity imposed by the *BCXN-loss*. Illustrative contour plots of the PDE residuals and the corresponding velocity contours at various training iterations are provided in Fig. 14 and Fig. 15.

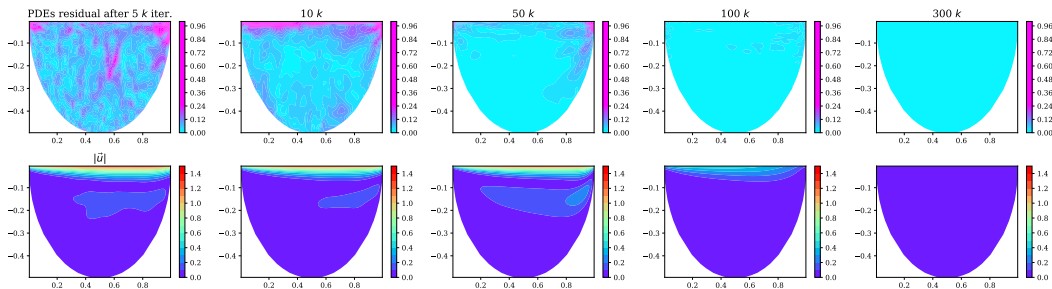

Figure 14: Contour plots of (Top) PDE error residual and (Bottom) velocity for *ND*-PINN.

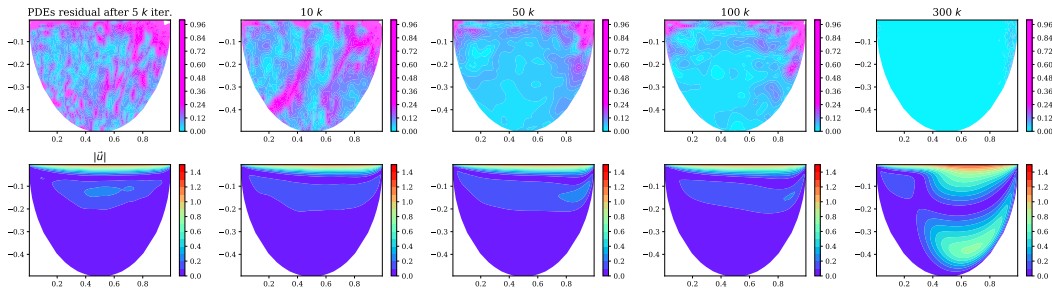

Figure 15: Contour plots of (Top) PDE error residual and (Bottom) velocity for *BCXN*-PINN.

As described in the previous sections, the ND-PINN typically fails to converge to a good model even after 300,000 training iterations, but this can be improved via the addition of a *BCXN-loss* during training. The velocity contours clearly illustrate the improved convergence. In addition, there is a striking difference in the distribution of the PDE residuals across the domain during the early stages of the training. In a typical ND-PINN, the PDE residuals rapidly reduce to zero, especially in the parts of the domain far away from the boundary condition. This also corresponds to the ND-PINN velocity profile converging to a wrong solution, and is indicative of the information from the boundary condition being lost within the domain. On the other hand, the PDE residuals within the domain persist for a much longer period when the *BCXN-loss* is included which is indicative of information from the top boundary being persistently transmitted to the inner domain until the PINN model has converged. This is also consistent with the physical dynamics of the system, as we note that the lid-driven cavity flow problem, as the name suggests, is heavily dependent on the top boundary's motion to drive the dynamical flow within the entire domain.

### A.9 SAMPLING EFFECT ON 2D LID-DRIVEN CAVITY FLOW

To illustrate the impact of having near-boundary stencil points outside of the domain, we further investigate the impact of two different sets of sampling points: i) Cell-centered (whereby near-boundary points need support points outside the domain) and ii) Node-centered (whereby near-boundary points use support points that are coincident with the domain boundary).

These two sets of sampling points are tested on a 2D lid-driven cavity flow at $Re = 1000$ with different training batch sizes and point sampling density for the *df-BCXN-CAN*, *CAN* and *AD* models, and are trained for $300,000$ iterations each. The model MSE and L2 relative errors are collated across 5 independent runs and tabulated in Table 7.

In the above table, cells are colored according to the model MSE at the end of 300,000 training iterations. Brown cells indicate scenarios under which no meaningful convergence was observed (with MSE > 1e-2). Yellow, green and blue cells indicate scenarios with increasingly good model performance, with each band indicating an order of magnitude improvement in MSE.

Table 7: Model Performance for Different Sampling Densities. [C] and [N] refer to cell-centered and node-centered sampling point arrangements.

| Batch Size | Points | Model | $50 \times 50$ | | $100 \times 100$ | |
|---|---|---|---|---|---|---|
| | | | MSE | Rel Error | MSE | Rel Error |
| 500 | [C] | df-BCXN-CAN | $2.63 \times 10^{-4}$ | $7.75 \times 10^{-2}$ | $3.06 \times 10^{-3}$ | $2.63 \times 10^{-1}$ |
| | | CAN & AD | $3.79 \times 10^{-2}$ | $9.30 \times 10^{-1}$ | $3.58 \times 10^{-2}$ | $8.98 \times 10^{-1}$ |
| | [N] | CAN | $4.14 \times 10^{-2}$ | $9.91 \times 10^{-1}$ | $3.86 \times 10^{-2}$ | $9.46 \times 10^{-1}$ |
| | | AD | $4.05 \times 10^{-2}$ | $9.80 \times 10^{-1}$ | $4.02 \times 10^{-2}$ | $9.66 \times 10^{-1}$ |
| 1000 | [C] | df-BCXN-CAN | $2.23 \times 10^{-4}$ | $7.13 \times 10^{-2}$ | $3.97 \times 10^{-4}$ | $9.47 \times 10^{-2}$ |
| | | CAN & AD | $3.65 \times 10^{-2}$ | $9.11 \times 10^{-1}$ | $3.36 \times 10^{-2}$ | $8.72 \times 10^{-1}$ |
| | [N] | CAN | $4.06 \times 10^{-2}$ | $9.82 \times 10^{-1}$ | $3.56 \times 10^{-2}$ | $9.08 \times 10^{-1}$ |
| | | AD | $4.16 \times 10^{-2}$ | $9.94 \times 10^{-1}$ | $3.60 \times 10^{-2}$ | $9.08 \times 10^{-1}$ |
| 2000 | [C] | df-BCXN-CAN | $2.36 \times 10^{-4}$ | $7.34 \times 10^{-2}$ | $7.14 \times 10^{-5}$ | $4.02 \times 10^{-2}$ |
| | | CAN & AD | $3.80 \times 10^{-2}$ | $9.31 \times 10^{-1}$ | $3.32 \times 10^{-2}$ | $8.65 \times 10^{-1}$ |
| | [N] | CAN | $7.41 \times 10^{-4}$ | $1.32 \times 10^{-1}$ | $3.36 \times 10^{-2}$ | $8.82 \times 10^{-1}$ |
| | | AD | $3.84 \times 10^{-2}$ | $9.53 \times 10^{-1}$ | $3.55 \times 10^{-2}$ | $9.02 \times 10^{-1}$ |
| 4000 | [C] | df-BCXN-CAN | | | $2.38 \times 10^{-5}$ | $2.31 \times 10^{-2}$ |
| | | CAN & AD | | | $3.23 \times 10^{-2}$ | $8.54 \times 10^{-1}$ |
| | [N] | CAN | | | $2.94 \times 10^{-4}$ | $8.24 \times 10^{-2}$ |
| | | AD | | | $3.50 \times 10^{-2}$ | $8.97 \times 10^{-1}$ |

From this experiment, we notice that a 4x increase in sampling density (from $50 \times 50$ to $100 \times 100$) can improve model performance, but this also requires the use of a larger batch size, which also add computational cost.

In addition, the node-centered *CAN*-PINN model has improved performance relative to the cell-centered *CAN*-PINN version. This further emphasizes our hypothesis that accurate modelling of the boundary conditions is an important factor in obtaining good PINN performance. As this is a square domain, the use of a sampling point distribution that has perfect connectivity with the boundary of the domain naturally leads to improved performance.

Interestingly though, the inclusion of our proposed *BCXN-loss* on a cell-centered model can improve model performance to achieve a lower error than the properly aligned baseline PINN model. This does suggest that the use of this proposed method can afford flexibility in the choice of sampling strategies, whereas conventional sampling strategies may still require sampling that takes into account the ability to maintain connectivity with the boundaries.

## A.10 EXPERIMENT WITH BCXN-PINN LOSS VARIANTS

As described in Section 4.2 and Section 4.3, there is no longer a need to explicitly evaluate the BC loss term for the *df-BCXN-loss* as the BCs are implicitly "infused" into the near-boundary domain points. We verify this point by comparing the results from a BCXN-PINN with the inclusion of BC loss term $L_{BC}$ into the *df-BCXN-loss*:

$$L_{PINN(df-BCXN_{w/BC})} = \lambda_{PDE} L_{PDE(df-BCXN)} + \lambda_{BC} L_{BC} \tag{16}$$

Based on our observation, the inclusion of BC loss term doesn't seem to benefit the BCXN-PINN with *df-BCXN-loss*. It is believed that the imposition of unnecessary BC loss term excessively complicates the training — given the fact that actual BCs are already implicitly defined and utilized during the computation of *df-BCXN-loss* — thereby slowing down convergence. We further incorporate the loss term from *sf-BCXN-loss* into Eq. (16):

$$L_{PINN(df\&sf-BCXN)} = \lambda_{PDE} L_{PDE(df-BCXN)} + \lambda_{BCXN} L_{BCXN} + \lambda_{BC} L_{BC} \tag{17}$$

to study the consequence of jointly using the *df* & *sf* approaches to modulate the BCXN-PINN training loss. Interestingly, the hybrid approach performs worse than the original *df-* and *sf-BCXN-*

*loss*. This may suggest that the BCXN-PINN training loss should only use either *df-* or *sf-BCXN-loss*, but not both together, potentially because of the conflicting base assumptions underlying both derivations.

## A.11    SENSITIVITY TO $\lambda_{BCXN}$ FOR FORWARD MODELS

As mentioned in Section 4.3, the BCXN weight $\lambda_{BCXN}$ in *sf-BCXN-loss* controls the extent of linear constraint being imposed at near boundary samples. We observe a noticeable improvement in solution quality with the appropriate tuning of $\lambda_{BCXN}$ for the BCXN-PINN in the 2D wavy channel flow ($\lambda_{BCXN} = 0.5$) and 3D bend tube channel flow ($\lambda_{BCXN} = 5$) test cases. The joint optimization results of $\lambda_{BCXN}$ and $\lambda_{DIV}$ for the 2D wavy channel flow problem are presented in Fig. 16. These results further highlight that an overly stringent imposition of a linear constraint underlying the BCXN-PINN methodology can adversely affect convergence without an accompanying increase in the emphasis ($\lambda_{DIV}$) placed on the divergence free condition.

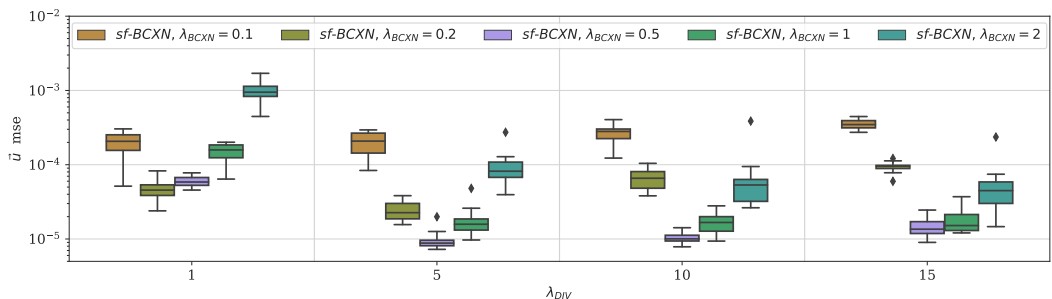

Figure 16: Distribution of $\vec{u}$ MSE between BCXN-PINN, when using different $\lambda_{BCXN}$ and $\lambda_{DIV}$ value in its *sf-BCXN-loss*, and ground truth solutions for the wavy channel flow problem, $Re = 100$.

## A.12    2D AND 3D INVERSE PROBLEM WITH MLP-ARCHITECTURE BCXN-PINN

We further demonstrate the proposed BCXN-PINNs on inverse problem, where the task is to infer the unknown physical quantities in the governing physics based on partially available observation data. In particular, we seek to infer the unknown *Re* in the incompressible N-S equations (Eq. (1)), as well as the solution over the problem domain, by training a PINN model with respect to the following data-constrained physics-informed loss functions:

$$L_{INV\_PINN} = L_{Data} + \lambda_{PDE} L_{PDE} + \lambda_{BC} L_{BC} \tag{18a}$$

$$L_{INV\_PINN(df-BCXN)} = L_{Data} + \lambda_{PDE} L_{PDE(df-BCXN)} \tag{18b}$$

$$L_{INV\_PINN(sf-BCXN)} = L_{Data} + \lambda_{PDE} L_{PDE} + \lambda_{BCXN} L_{BCXN} + \lambda_{BC} L_{BC} \tag{18c}$$

In addition to the known BCs, these loss functions include additional data loss component:

$$L_{Data} = \frac{1}{n_{obs}} \Sigma_{i=1}^{n_{obs}} \left( \vec{u}_i - \vec{u}_i(\vec{x}_i; \boldsymbol{w}) \right)^2 \tag{19}$$

to match the PINN output $\vec{u}(\vec{x}; \boldsymbol{w})$ against target $\vec{u}$ over $n_{obs}$ observations.

**Inverse 2D semi-circle lid-driven cavity problem.** We assume the availability of limited observations ($n_{obs} = 10$ or $20$) of velocity $\vec{u}$. The experiment comprises of 10 independent runs, where observations are randomly drawn (i.e. obtained from the simulation). Selected observation sets are displayed in Fig. 17a. Besides these observations, the PINN models are trained on 3,930 equidistantly spaced samples for evaluating the PDE loss and 300 BC samples. The inverse modelling results are visualized in Fig. 17b, comprising the distribution of $\vec{u}$ MSE and inferred $Re^{-1}$ for both $n_{obs} = 10$ or $n_{obs} = 20$ scenarios. With $n_{obs} = 20$, our inverse BCXN-PINN model consistently infers an accurate *Re* which is always within 10% error from the ground truth ($Re = 1000$) and achieves $\vec{u}$ MSE below 5e-5. The efficacy of *df-BCXN-loss* drops slightly for the more challenging $n_{obs} = 10$ scenario. However, for the *sf-BCXN-loss*, all inferred *Re* values are still within 15% of the ground

truth, and their solutions still achieve a $\vec{u}$ MSE below 5e-5. Fig. 17b also shows that baseline PINNs using *ND-* and *AD-loss* can infer an accurate *Re* in this inverse problem. This may suggest that, once a PINN model is able to correctly infer the unknown *Re*, it is much easier to learn the solution with these additional observations.

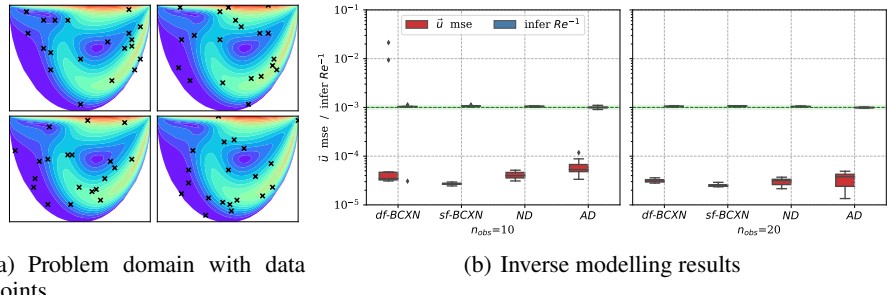

(a) Problem domain with data points

(b) Inverse modelling results

Figure 17: (a) BCXN-PINN is applied to simultaneously infer the unknown *Re* coefficient (ground truth $Re = 1000$) in the incompressible N-S equations and to solve for the solution based on limited observations (x-marked, $n_{obs} = 10$), for inverse 2D semi-circle lid driven cavity problem. (b) The distribution of $\vec{u}$ MSE and inferred $Re^{-1}$ obtained from the inverse PINNs at $n_{obs} = 10, 20$. The green dashed lines and shaded areas indicate the ground truth $Re^{-1}$ and their 10% error bounds.

**Inverse 3D bend tube channel flow problem.** We assume limited observations of velocity $\vec{u}$ ($n_{obs} = 50, 100$) can be probed near the bending area Fig. 18a. Besides these observations, we train PINN models on 19,786 equidistantly spaced PDE samples and 1,632 BC samples, for inferring the unknown *Re* (ground truth $Re = 100$) and predicting the solution near the bending area. We only apply the non-slip BC at the wall boundary since the inlet and outlet conditions are assumed to be unknown. The experiment comprises of 10 independent runs, where observations are randomly drawn (i.e., obtained from the simulation). The inverse modelling results from different PINN models are visualized in Fig. 18b. Again, we observe good performance from all tested PINN models, with their inferred *Re* always within 15% error from the ground truth. They can achieve below 2e-4 $\vec{u}$ MSE with $n_{obs} = 50$, and even better accuracy (i.e., <1e-4) with $n_{obs} = 100$. Nonetheless, the experiment demonstrates the effectiveness of BCXN-PINN for solving complex inverse problems from limited random observations, with consistent performance across different sets of random data.

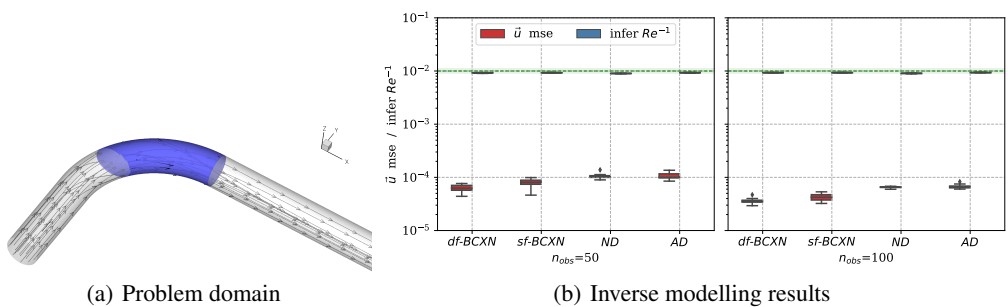

(a) Problem domain

(b) Inverse modelling results

Figure 18: (a) The problem domain of the inverse 3D bend tube flow problem for inferring the unknown *Re* coefficient (ground truth $Re = 100$) in the incompressible N-S equations and solving for the solution based on limited observations. (b) The distribution of $\vec{u}$ MSE and inferred $Re^{-1}$ obtained from the inverse PINNs at $n_{obs} = 50, 100$. The green dashed lines and shaded areas indicate the ground truth $Re^{-1}$ and their 10% error bounds.

**Hyper-parameter tuning.** We use a same MLP architecture, ND scheme, and batch size settings as summarized in Appendix A.3 Table 1, for the inverse problem test cases. Each batch contain at most 50 observations for computing the data loss. To ensure sufficiently accurate fit to the observations,

we reweight the PINN loss terms (i.e., the weight of data loss to PDE loss is 10 to 1) to give more priority to the data loss. The current inverse problem test cases require less training iterations to converge, as compared to the pure physics-informed learning encountered in the forward problem. The max. iteration is set as 200,000 with an initial learning rate of 5e-4 for both test cases. We do not apply additional linear constraints to the near boundary samples if they are part of the observation set.

## A.13 VISUALIZATION OF CNN-ARCHITECTURE RESULTS

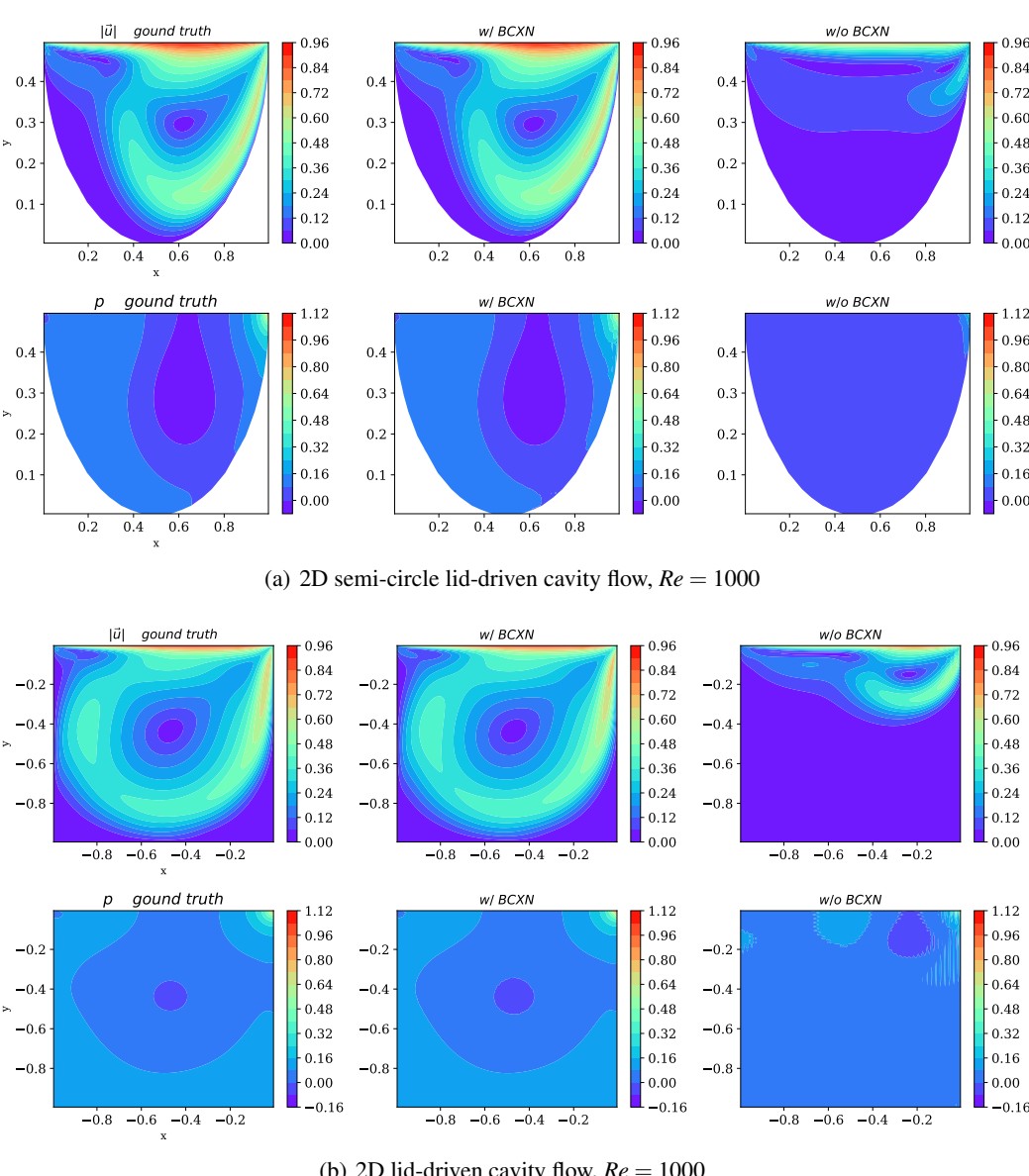

(a) 2D semi-circle lid-driven cavity flow, $Re = 1000$

(b) 2D lid-driven cavity flow, $Re = 1000$

Figure 19: Comparison of velocity magnitude $\|\vec{u}\|$ and pressure $p$ contour between the ground truth and PINN solutions (median MSE from 10 runs) for the (a) semi-circle lid-driven cavity flow problem, $Re = 1000$, and (b) lid-driven cavity flow problem, $Re = 1000$, using CNN architecture.

### A.14 ADDITIONAL FORWARD PROBLEM ON 2D TAYLOR-COUETTE FLOW WITH DIRICHLET AND NEUMANN TYPE BOUNDARY CONDITION

As a demonstration of the generalizability of the current proposed method to different types of boundary conditions based on the derivation in Section A.1, we further apply the described method to the solution of a 2D Taylor-Couette problem.

This problem describes the flow in a fluid domain between 2 concentric cylinders. The inner cylinder is of radius $R_1 = 0.45$ while the outer cylinder has radius $R_2 = 0.96$.

The analytical solution for the problem is known to be:

$$u(x,y) = -K((\frac{R_2}{r})^2 - 1)y \tag{20}$$

$$v(x,y) = K((\frac{R_2}{r})^2 - 1)x \tag{21}$$

$$p(x,y) = K^2(\frac{r^2}{2} - \frac{R_2^4}{2r^2} - R_2^2 log(r^2)) \tag{22}$$

where angular velocity $\omega$ is set as 1, $K = \frac{\omega R_1^2}{R_2^2 - R_1^2}$ and $r = \sqrt{x^2 - y^2}$.

The numerical experiments are performed with $Re = 1000$ for four different pairs of boundary condition settings for the outer (inner) cylinder: i) Dirichlet (Dirichlet); ii) Dirichlet (Neumann); iii) Neumann (Dirichlet); and iv) Neumann (Neumann).

For all the boundary setting pairs, the training is performed with a CNN-type PINN architecture on a 56x56 voxel domain and $\Delta x = \Delta y = 2/50$. Additional PINN model details are described in Table 4. Contour plots of the results are presented in Fig. 20 and model errors are presented in Table 8.

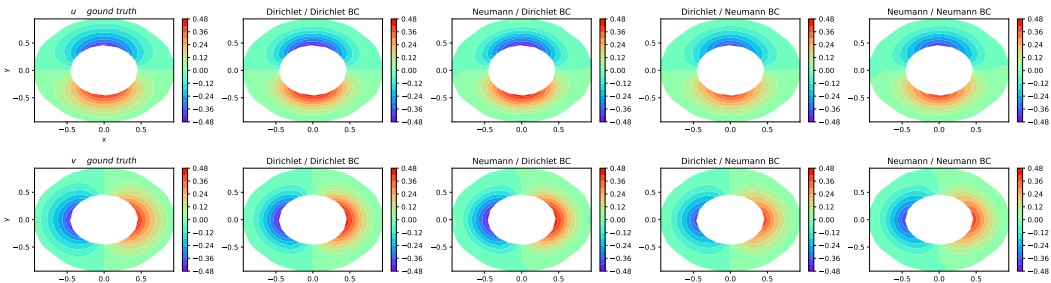

Figure 20: Velocity contours of 2D Taylor-Couette flow problem with different boundary condition settings.

Table 8: Model Performance with Different [D]irichlet or [N]eumann Boundary Condition Settings

| Metric | Variable | Outer / Inner Cylinder BC Type | | | |
|---|---|---|---|---|---|
| | | D / D | N / D | D / N | N / N |
| MSE | $u$ | $2.72 \times 10^{-4}$ | $4.22 \times 10^{-4}$ | $5.22 \times 10^{-4}$ | $3.92 \times 10^{-4}$ |
| | $v$ | $2.73 \times 10^{-4}$ | $4.16 \times 10^{-4}$ | $5.22 \times 10^{-4}$ | $4.12 \times 10^{-4}$ |
| Relative Error | $u$ | 0.115 | 0.143 | 0.159 | 0.138 |
| | $v$ | 0.115 | 0.142 | 0.159 | 0.142 |

From the results, it can be seen that the *BCXN-loss* can be flexibly applied to problems with any combination of Dirichlet and Neumann type boundary conditions, although the majority of presented examples in this work feature Dirichlet boundary conditions.

### A.15    ADDITIONAL FORWARD PROBLEM ON TRANSIENT 2D SEMI-CIRCLE LID-DRIVEN CAVITY FLOW, $Re = 500$

In addition to the forward modelling of steady-state 2D semi-circle lid-driven cavity flow, we further evaluate the improvement obtained from application of CNN-architecture BCXN-PINN to the modelling of a transient version of this problem. The problem geometry and set-up is consistent with the problem description in Section 5.2. However, the problem's initial condition corresponds to that of a fluid domain at rest (at t = 0). The PINN model is then used to model the evolution of the flow within the domain from $t = 0$ to $t = 2$. Additional model settings are provided in Table 4.

The average errors for all time points between $t = 0$ and $t = 2.0$ yield MSEs of 3.95e-4 and 9.66e-3 with and without the inclusion of *BCXN-loss* respectively. This also corresponds to relative L2 errors of 0.129 and 0.681 respectively. Hence, the inclusion of *BCXN-loss* in this transient problem can improve MSE by more than 1 order of magnitude.

Velocity contour plots of the PINN models trained with and without *BCXN-loss* are provided in Fig. 21 to illustrate the differences in model convergence.

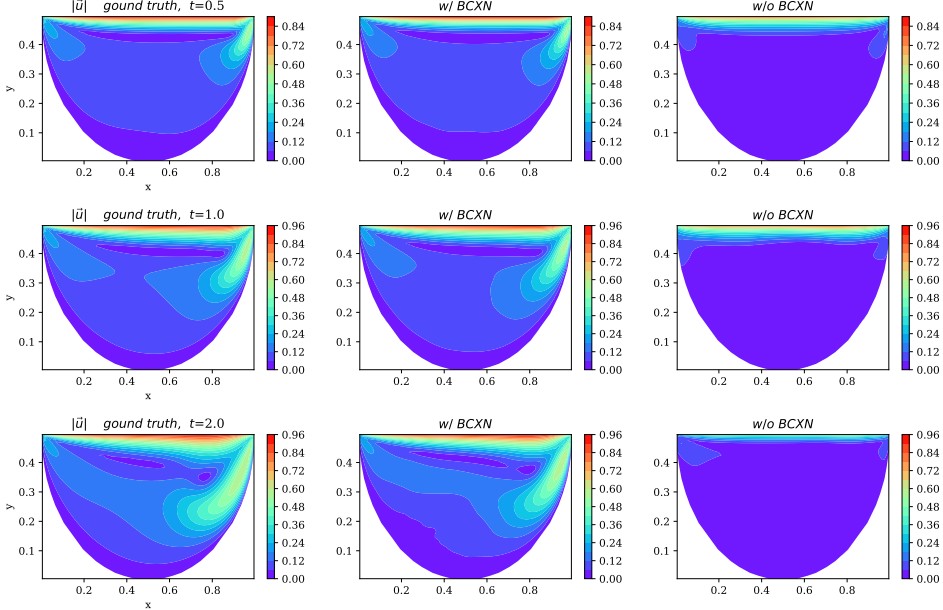

Figure 21: Velocity contours of the transient simulation at $t = 0.5$, $t = 1.0$ and $t = 2.0$ for both the BCXN-PINN and ND-PINN implementation. The velocity contours on the left, middle and right correspond to the results from CFD, BCXN-PINN and ND-PINN respectively. The top, middle and bottom rows are for $t = 0.5$, $t = 1.0$, and $t = 2.0$.

### A.16    CNN-ARCHITECTURE BCXN-PINN AND BASELINE PINN PERFORMANCE ON A FORWARD PROBLEM WITH INCREASING COMPLEXITY

For easy reference, we summarize the average MSE and relative L2 error obtained for the 2D semi-circle lid-driven cavity experiments across 5 independent runs. These results indicate the single model errors obtained using the settings described in Table 4. Plots of the velocity MSEs are provided in Fig. 22. We can clearly see that the performance difference is most pronounced for more complex boundary value problems. As Re increases, the complexity of the developed flow typically requires more sophisticated algorithms for better capture of the non-linear effects.

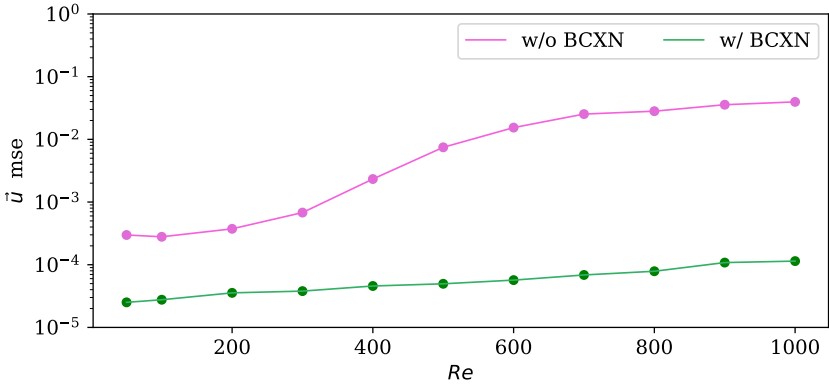

Figure 22: Plot of velocity MSEs at different Re for both the *df-BCXN-* and *ND-* type CNN-architecture PINN.

Additional quantitative comparisons at select Reynolds numbers are presented in Table 9 below for easier reference.

Table 9: Model Performance for the 2D Semi-circular Lid-Driven Cavity Problem

| Metric | CNN-Model | Reynolds Number | | | | |
|---|---|---|---|---|---|---|
| | | 50 | 100 | 200 | 400 | 1000 |
| MSE | w/ BCXN | $2.51 \times 10^{-5}$ | $2.76 \times 10^{-5}$ | $3.56 \times 10^{-5}$ | $4.58 \times 10^{-5}$ | $1.14 \times 10^{-4}$ |
| | w/o BCXN | $2.98 \times 10^{-4}$ | $2.79 \times 10^{-4}$ | $3.74 \times 10^{-4}$ | $2.33 \times 10^{-3}$ | $3.96 \times 10^{-2}$ |
| Relative Error | w/ BCXN | 0.0258 | 0.0269 | 0.0305 | 0.0335 | 0.0492 |
| | w/o BCXN | 0.0823 | 0.0732 | 0.0780 | 0.211 | 0.855 |

Importantly, we note that although the current proposed BCXN-PINN method may nominally involve more computation, the impact on training time is under 5% in our experiments. The time taken for training a single *df-BCXN-loss* PINN is about 126-136 minutes while the time taken for training the equivalent *ND*-PINN with similar model size and optimization parameters is 121-137 minutes. This thus further emphasizes the utility of the current proposed method. A more in-depth description is provided in Appendix Section A.17.

## A.17   IMPACT OF *BCXN*-LOSS ON TRAINING TIME

We further compare the time taken for training a model with and without the *BCXN-loss* on the MLP-architecture and CNN-architecture models as applied to the 2D semi-circle lid-driven cavity problem at $Re = 1000$. The average MSE and relative L2 error obtained are averaged across 10 independent runs, and the minimum and maximum wall-clock time are recorded. All experiments are run with the standard Tensorflow and Keras implementation on a 20-core workstation with an Intel Xeon Gold 6248 processor and an Nvidia RTX 2080 Ti GPU.

Quantitative comparisons for the MLP-architecture models are presented in Table 10. All models were run for an identical number of training iterations (300,000) with the same batch size (1,000) and had the same model size to facilitate a fair comparison.

Quantitative comparisons for the CNN-architecture model are also presented in Table 11. In additional, both models were run for an identical number of training iterations (200,000) and had the same model size to facilitate a fair comparison.

While not a comprehensive study, we note that the inclusion of the *BCXN-loss* across all the cases only slightly increases the amount of time required for model training. This was on the order of less

Table 10: Model Performance for a MLP-architecture PINN

| Model | MSE | Rel. Error | Time (min) |
|---|---|---|---|
| *df-BCXN-CAN* | $6.21 \times 10^{-5}$ | $3.55 \times 10^{-2}$ | $55 - 57$ |
| *sf-BCXN-CAN* | $5.65 \times 10^{-5}$ | $3.45 \times 10^{-2}$ | $41 - 42$ |
| *CAN* | $2.82 \times 10^{-2}$ | $7.62 \times 10^{-1}$ | $40 - 46$ |
| *df-BCXN-ND* | $1.61 \times 10^{-4}$ | $5.95 \times 10^{-2}$ | $23 - 24$ |
| *sf-BCXN-ND* | $2.68 \times 10^{-4}$ | $7.54 \times 10^{-2}$ | $22 - 23$ |
| *ND* | $3.91 \times 10^{-2}$ | $8.56 \times 10^{-1}$ | $20 - 22$ |
| *AD (SIREN)* | $1.74 \times 10^{-2}$ | $5.95 \times 10^{-1}$ | $58 - 61$ |
| *AD* (Vanilla) | $5.46 \times 10^{-2}$ | $9.83 \times 10^{-1}$ | $50 - 55$ |

Table 11: Model Performance for a CNN-architecture PINN

| Model | MSE | Rel. Error | Time (min) |
|---|---|---|---|
| *df-BCXN* | $1.14 \times 10^{-4}$ | $4.92 \times 10^{-2}$ | $126 - 136$ |
| *ND* | $3.96 \times 10^{-2}$ | $8.55 \times 10^{-1}$ | $121 - 137$ |

than 5 % for the CNN-type architecture and the ND-MLP-type model. Similarly, the *sf-BCXN-CAN* had negligible increase in training time relative to the baseline *CAN*-PINN model.

However, it is worth noting that while the time taken may have increased slightly, the MSE typically decreases more than 1 order of magnitude, suggesting that this trade-off between model training time and performance with the inclusion of a *BCXN-loss* may still be preferred.

## A.18 SUMMARY OF BCXN-PINN AND BASELINE PINN PERFORMANCE

For easy reference, we summarize the BCXN-PINN's model performance and compare them to other baseline methods previously published. The average MSE and relative L2 error across multiple independent runs are recorded in Table 12 and Table 13 for both the MLP-architecture and CNN-architecture models respectively.

Table 12: Model Performance Comparisons for MLP-architecture PINNs

| Problem | Model | MSE | Rel. Error |
|---|---|---|---|
| 2D semi-circle lid-driven cavity, $Re = 1000$ | *df-BCXN-CAN* | $6.21 \times 10^{-5}$ | $3.55 \times 10^{-2}$ |
| | *sf-BCXN-CAN* | $5.65 \times 10^{-5}$ | $3.45 \times 10^{-2}$ |
| | *CAN* | $2.82 \times 10^{-2}$ | $7.62 \times 10^{-1}$ |
| | *df-BCXN-ND* | $1.61 \times 10^{-4}$ | $5.95 \times 10^{-2}$ |
| | *sf-BCXN-ND* | $2.68 \times 10^{-4}$ | $7.54 \times 10^{-2}$ |
| | *ND* | $3.91 \times 10^{-2}$ | $8.56 \times 10^{-1}$ |
| | *AD (SIREN)* | $1.74 \times 10^{-2}$ | $5.95 \times 10^{-1}$ |
| | *AD* (Vanilla) | $5.46 \times 10^{-2}$ | $9.83 \times 10^{-1}$ |
| 2D lid-driven cavity, $Re = 1000$ | *df-BCXN-CAN* | $5.10 \times 10^{-5}$ | $3.38 \times 10^{-2}$ |
| | *sf-BCXN-CAN* | $6.10 \times 10^{-5}$ | $3.72 \times 10^{-2}$ |
| | *CAN* | $4.23 \times 10^{-2}$ | $9.77 \times 10^{-1}$ |
| | *AD (SIREN)* | $3.47 \times 10^{-2}$ | $8.87 \times 10^{-1}$ |
| 2D wavy channel flow, $Re = 100$ | *df-BCXN-CAN* | $1.34 \times 10^{-4}$ | $7.28 \times 10^{-2}$ |
| | *sf-BCXN-CAN* | $9.00 \times 10^{-6}$ | $4.09 \times 10^{-2}$ |
| | *CAN* | $5.96 \times 10^{-1}$ | $2.55 \times 10^{0}$ |
| | *AD (SIREN)* | $3.52 \times 10^{-2}$ | $7.19 \times 10^{-1}$ |
| 3D bend tube channel flow, $Re = 100$ | *df-BCXN-ND* | $8.09 \times 10^{-4}$ | $1.03 \times 10^{-1}$ |
| | *sf-BCXN-ND* | $8.59 \times 10^{-4}$ | $9.50 \times 10^{-2}$ |
| | *ND* | $2.63 \times 10^{-1}$ | $6.89 \times 10^{-1}$ |
| | *AD (SIREN)* | $2.65 \times 10^{-1}$ | $6.94 \times 10^{-1}$ |
| 2D flow past airfoil, $Re = 500$ | *sf-BCXN-CAN* | $4.65 \times 10^{-5}$ | $8.10 \times 10^{-2}$ |
| | *CAN* | $3.32 \times 10^{-4}$ | $1.92 \times 10^{-1}$ |
| | *sf-BCXN-ND* | $5.03 \times 10^{-5}$ | $6.93 \times 10^{-2}$ |
| | *ND* | $3.54 \times 10^{-4}$ | $1.84 \times 10^{-1}$ |
| | *AD (SIREN)* | $1.18 \times 10^{-3}$ | $3.17 \times 10^{-1}$ |

(I) Implementation for *CAN*-PINN models based on previous work (Chiu et al., 2022).

(II) Implementation for *SIREN*-PINN models based on previous work (Sitzmann et al., 2020; Wang et al., 2021b; Wong et al., 2022).

(III) Implementation for vanilla PINN models based on previous work (Raissi et al., 2019).

Table 13: Model Performance Comparisons for CNN-architecture PINNs

| Problem | Model | MSE | Rel. Error |
|---|---|---|---|
| 2D semi-circle lid-driven cavity, $Re = 1000$ | *df-BCXN* | $1.14 \times 10^{-4}$ | $4.92 \times 10^{-2}$ |
| | *ND* | $3.96 \times 10^{-2}$ | $8.55 \times 10^{-1}$ |
| 2D lid-driven cavity, $Re = 1000$ | *df-BCXN* | $2.52 \times 10^{-5}$ | $4.46 \times 10^{-2}$ |
| | *ND* | $2.38 \times 10^{-2}$ | $1.00 \times 10^{0}$ |
| Transient 2D semi-circle lid-driven cavity, $Re = 500$ | *df-BCXN* | $3.95 \times 10^{-4}$ | $1.29 \times 10^{-1}$ |
| | *ND* | $9.66 \times 10^{-3}$ | $6.81 \times 10^{-1}$ |

