# OpenReview forum: "Fast-PINN for Complex Geometry: Solving PDEs with Boundary Connectivity Loss"
_ICLR.cc/2023/Conference — Submitted to ICLR 2023_

### Official Review · Reviewer_Dt9j · 2022-10-21

**Confidence:** 5
**Clarity, Quality, Novelty And Reproducibility:** The written is clear.
**Correctness:** 2
**Technical Novelty And Significance:** 2
**Empirical Novelty And Significance:** Not applicable
**Recommendation:** 5

**Strength And Weaknesses:**

Strengths
- The proposed method estimates the external points and thus avoid the issue of stencil points falling outside the geomery.

Weaknesses
Comments on the methods:
- The method only works for Dirichlet BC, but not other BCs, such as Neumann BC, Robin BC, etc.
- The method uses a linear interpolation to compute external points, which will limit the accuracy.
- The name “Fast-PINN” is not proper. The focus of the proposed method is complex geometry. It is not a fast version of PINN for any PDE problems with any BCs.

Comments on the correctness of the numerical results:

- 1D convection-diffusion PDE has been solved very accuracy in many PINN papers, so it is questionable if the AD and ND results are correct. No details about the setup of Fig 2 are provided.
- In the 2D cavity flow, it is regular geometry, and all the points can be chosen inside the domain. In this case, the proposed method becomes a standard PINN, so the results of ND and AD don’t seem correct. Also, there are PINN papers with 2D cavity flow problem and PINN works well. The only explanation is that the authors make ND and AD become bad.

Other comments on the results:

- The paper uses MSE error, but other metrics should be provided, such as L2 relative error.
- The paper only considered steady-state problems, but no time dependent problems are tested.

Comments on the comparisons:

- It is unfair to compare AD/ND with BCXN as BCXN is more expensive. The authors should increase the training points of AD/ND such that the computational cost is similar, and then compare the accuracy.
- There are many other methods, e.g., adaptive sampling methods, as were discussed in Section 2, but no comparison is provided.

Other comments:

- The section 3 is “THEORY”, but there is no theory in the section.

**Summary Of The Paper:**

The paper proposed a new Boundary Connectivity (BCXN) loss function for PINNs for solving PDEs with complex geometry. BCXN uses a linear interpolation to compute the points outside the domain to address the issue that the derivative stencil points are outside the domain, and then the estimated external points can be enforced directly in the loss or via a penalty loss. Numerical experiments showed that BCXN achieves better accuracy with less training points.

**Summary Of The Review:**

The proposed method doesn’t work as well as they claimed. It has many limitations. The numerical experiments are not convincing enough.

---

> ### Author Response · Authors · 2022-11-19
> **Authors Response to Comments on Methods & Numerical Results**
>
> We thank the reviewer for the questions about potential limitations in the method and have taken the opportunity to clarify these points in our revised manuscript.
>
> > The method only works for Dirichlet BC, but not other BCs, such as Neumann BC, Robin BC, etc.
>
> Due to limitations in the manuscript length, the original manuscript focused on describing how this idea can be applied to Dirichlet boundary conditions. However, this methodology can be flexibly extended to other boundary conditions such as Neumann and Robin type boundary conditions. We have further included a description of how this method can be extended to other boundary conditions besides Dirichlet conditions in **Appendix A1** to elaborate on this point. In addition, we have also conducted experiments on a 2D Taylor-Couette flow where we can flexibly employ a combination of Dirichlet and Neumann boundary conditions. Results are included in the **Appendix A14**.
>
> > The method uses a linear interpolation to compute external points, which will limit the accuracy.
>
> As a proof-of-concept for the method, we have chosen to focus on linear interpolation as a means to model the connection to the external points. The interpolation function can be extended to more complicated choices such as quadratic splines and even higher-order functions. Nonetheless, usage of a linear function in the current implementation seems to already be advantageous relative to the baseline, although we agree with the reviewer that a linear interpolation may indeed limit the accuracy in certain instances.
>
> > The name “Fast-PINN” is not proper. The focus of the proposed method is complex geometry. It is not a fast version of PINN for any PDE problems with any BCs.
>
> We agree that the proposed method is not a fast version of PINN for any PDE problems, hence the name “Fast-PINN” may not be the most proper one. We have changed our title to **BCXN-PINN for Complex Geometry: Solving PDEs with Boundary Connectivity Loss** to further emphasize the focus of this work is the enhanced boundary connectivity (BCXN).
>
> > 1D convection-diffusion PDE has been solved very accuracy in many PINN papers, so it is questionable if the AD and ND results are correct. No details about the setup of Fig 2 are provided.
>
> We definitely agree with the reviewer that many papers have demonstrated good accuracy in the solving of a 1D convection-diffusion PDE, and apologize for not including details about the setup in Fig 2. Details have been added to **Appendix A3** Table 3. In this instance, Fig. 2 was meant as an illustration of the potential difficulty in PINN training when the boundary condition is not properly maintained, and a more difficult problem setting with respect to the sampling density was selected. In particular, we note that while the 1D convection-diffusion PDE has been solved in previous work, different systems parameters such as the Peclet number can result in increasingly sharp steady state profiles and a consequently more difficult modelling problem. As previously mentioned in other work in literature, sampling strategies can be an alternate route to improving the convergence of PINN models in these relatively more difficult problems.
>
> > In the 2D cavity flow, it is regular geometry, and all the points can be chosen inside the domain. In this case, the proposed method becomes a standard PINN, so the results of ND and AD don’t seem correct. Also, there are PINN papers with 2D cavity flow problem and PINN works well. The only explanation is that the authors make ND and AD become bad.
>
> While there are PINN papers with 2D cavity flow problem, system parameters for the same problem can greatly impact the difficulty of the problem. For example, an increasing *Re* can greatly increase the modelling difficulty of what is nominally a problem with the same geometry and boundary conditions as illustrated in **Appendix A16**. Additional nonlinear effects such as eddies emerge at higher *Re*, and it is also known that higher Re typically require more sophisticated numerical solvers for fast convergence relative to lower *Re*. Similarly, we note that the discrepancy in performance between the PINN models with and without BCXN-loss truly emerges at higher *Re*.
>
> In addition, for this particular problem, we apologize to the reviewer for not including necessary additional details in our problem description. We mentioned in the original manuscript that sampling points were initially chosen to allow for the near-boundary points to extend outside of the domain to illustrate the impact of the proposed BCXN-loss. Additional problem setup details are now provided in **Appendix A9** to illustrate this comparison. Compiled results in Table 7 illustrate our hypothesis that while the typical ND-type PINN can solve the lid-driven cavity scenario under certain denser sampling settings, the inclusion of BCXN-loss can greatly improve the model convergence and reduce the need for as dense sampling.

---

> ### Author Response · Authors · 2022-11-19
> **Authors Response to Other Comments on Results & Comparisons**
>
> > The paper uses MSE error, but other metrics should be provided, such as L2 relative error.
>
> We thank the reviewer for this comment and have included the L2 relative error in additional tables in the **Appendix A18**.
>
> > The paper only considered steady-state problems, but no time dependent problems are tested.
>
> We agree with the reviewer that it will be interesting to evaluate the performance on transient problems in addition to the steady-state problems previously presented. We have also studied the ability of this method to learn the time-varying dynamics in a 2D semi-circle lid-driven cavity problem, starting from a fluid domain that is initially at rest. The incorporation of a BCXN-loss similarly improves the model performance by an order of magnitude. This result has now been included in the revised manuscript in **Appendix A15**.
>
> > It is unfair to compare AD/ND with BCXN as BCXN is more expensive. The authors should increase the training points of AD/ND such that the computational cost is similar, and then compare the accuracy.
>
> We appreciate the reviewer’s point that the BCXN method is nominally computationally more expensive per iteration. We have monitored the wall-clock time required for the respective methods to complete the same number of training iterations on the exact same workstation for a model test problem and report the timing results in **Appendix A17**. Empirically, we notice that the inclusion of BCXN does not substantially increase the training time for the CNN-type architecture (<5%) and that the greater disparity arises from the underlying PINN method (AD vs ND vs CAN). In particular, the empirical results seem to suggest that the increased computational cost is fairly minor. In addition, referencing the results in **Appendix A8** Table 7, we note that the inclusion of BCXN enabled models to achieve similar MSE to the baseline models with 4x less sampling points and 8x less batch size. The additional gains from reduced batch size and sampling points far outweigh our observed slight increases in wall-clock run time for model training.
>
> > There are many other methods, e.g., adaptive sampling methods, as were discussed in Section 2, but no comparison is provided.
>
> We thank the reviewer for raising this point. As shared with the other reviewer, we chose to compare different variants of the PDE loss as a baseline in this work and have summarized the results accordingly in the **Appendix A18**. Many of the other methods discussed, including various innovative sampling strategies [1-4], tend to focus on MLP-type automatic differentiation-based PINNs. While adaptive (re)-sampling strategies can be helpful, we believe this approach also entails further development to address other practical settings such as (re)-balancing of the loss function as the high/low residual points are chosen, etc. This makes a direct comparison difficult to define. In addition, adaptive or residual-based sampling might be more intuitively suited to MLP-type PINN implementations relative to CNN-type implementations which we also study in this work. For example, the CNN-type PINN mentioned in our lid-driven cavity meta-model is not amenable to variations in sampling as it operates on a pre-defined voxel-type system, thereby precluding a direct comparison to adaptive sampling strategies. Lastly, we believe the current proposed method is potentially complementary to other sophisticated sampling strategies and that both approaches are not exclusive. Hence, for the purposes of this work, we believe it would be beneficial to focus on showing the intuition and results in isolation relative to our chosen baseline.
>
> 1. Elhamod et al. "CoPhy-PGNN: Learning physics-guided neural networks with competing loss functions for solving eigenvalue problems." (2020).
> 2. Wang et al. "Understanding and mitigating gradient flow pathologies in physics-informed neural networks." (2021).
> 3. Wang et al. "When and why PINNs fail to train: A neural tangent kernel perspective." (2022).
> 4. Jin, Xiaowei, et al. "NSFnets (Navier-Stokes flow nets): Physics-informed neural networks for the incompressible Navier-Stokes equations." (2021).
>
> > The section 3 is “THEORY”, but there is no theory in the section.
>
> We thank the reviewer for this comment and have revised the manuscript to use a section header of **Preliminary** for clarity.

---

> ### Author Response · Authors · 2022-11-19
> **Authors Response to Summary of The Review**
>
> > The proposed method doesn’t work as well as they claimed. It has many limitations. The numerical experiments are not convincing enough.
>
> We thank the reviewer for the critical reading of the manuscript and hope we have adequately addressed the concerns. We have included more details about the methods and experiments as requested to ensure reproducibility and substantial additional numerical experiments to better illustrate the potential benefit of the proposed method across multiple problem and PINN setups, across different error metrics such as MSE and L2 relative error. We agree that the proposed method has limitations. Nonetheless, we believe that our extensive numerical experiments (comprising 4 fluid dynamical case studies in both 2D and 3D, 1 aerofoil geometry study, 1 multi-physics case study, 1 Neumann boundary condition study and 1 transient problem case study) span a sufficiently large variety to demonstrate the viability and promise of this proposed method.

---

> > ### Comment · Reviewer_Dt9j · 2022-12-07
> > **Response to Authors**
> >
> > The paper quality has been improved. I have increased my score.

---

### Official Review · Reviewer_KS1E · 2022-10-24

**Confidence:** 4
**Correctness:** 4
**Technical Novelty And Significance:** 3
**Empirical Novelty And Significance:** 3
**Recommendation:** 6

**Clarity, Quality, Novelty And Reproducibility:**

The paper is easy to follow and clearly written. The loss function introduced seems to be novel and achieves good performance.

**Strength And Weaknesses:**

I really enjoyed reading the paper. The loss function mentioned is easy to implement and gets good results when compared to baseline PINNs. The empirical section is clear and easy to read and the authors do a great job at delineating the type overall setup of the PDEs they consider for their experimental section. The PDEs consider mainly differ in the complexity of the boundaries of the domains.

The improvement introduced by fast-PINNs seems to be quite significant compared to the normal PINNs (both in terms of the rate of convergence and final loss). The authors also point out certain inductive biases that are inherent in CNN based architectures, which is quite intriguing.


One small question for the authors: Is there a reason that the authors on solving focussed fluid dynamics with PINNs, do the gains follow for other PDEs as well (like reaction diffusion etc?)

**Summary Of The Paper:**

This paper addresses the issue of PINNs overfitting at the boundary of the domain. Since numerical differentiation based methods require points outside the boundary of the domain to work well, the authors propose a simple and efficient solution which constraints the value of the function outside the boundary (external stencil points) in terms of “mirror points” that lay inside the domain.

This loss can be directly enforced (i.e., the equality can imposed directly for the stencil points outside the boundary) or can be enforced softly by enforcing it as a loss function.

The authors empirically compare the performance of Fast-PINN with normal PINNs (using numerical and autodiff loss) under 4 fluid dynamics based cases, and show Fash PINNs converge faster and have a lower mse loss.

**Summary Of The Review:**

The authors provide a simple loss function take care of the points that lie outside the domain for numerical differentiation type loss for PDE solvers. The solution provided by the authors is simple to implement and shows improvement over normal PINN methods.

---

> ### Author Response · Authors · 2022-11-19
> **Authors Response**
>
> We thank the reviewer for the positive comments.
>
> > One small question for the authors: Is there a reason that the authors on solving focussed fluid dynamics with PINNs, do the gains follow for other PDEs as well (like reaction diffusion etc?)
>
> As the reviewer noted, the improvements observed in our problems seem significant, but the demonstrations are primarily limited to fluid dynamic problems. The method, like PINNs in general, can be applied to other PDEs, and the gains are related to better modelling of the complex boundaries, agnostic of the PDE. However, the fluid dynamics problem are chosen for demonstration because they are the authors’ primary interests. In addition, we believe that the non-linearity and complexity inherent in fluid dynamics (Navier-Stokes) may better illustrate the gains from improved modelling of the boundary condition.
>
> However, we have added an extra example in **Appendix A7** to illustrate how this method can be applied to such multi-physics problems comprising both fluid dynamics and heat transfer, and we continue to observe improvements. While we believe the method should generally benefit all kinds of PDE problems, we do agree with the reviewer that the gains may be variable with the complexity of the PDE (e.g. degree of non-linearity) and the problem itself. For example, linear PDEs such as heat transfer problems may see slightly less gains from more accurate connectivity between the boundary and the inner domain, but this will be an interesting additional direction to delve deeper into in the future.

---

> > ### Comment · Reviewer_KS1E · 2022-11-28
> > **Thank you for the response**
> >
> > Thank you for the response, and also adding further experiments on different PDEs. They have answered most of my questions.

---

### Official Review · Reviewer_3A2P · 2022-10-25

**Confidence:** 3
**Correctness:** 4
**Technical Novelty And Significance:** 3
**Empirical Novelty And Significance:** 3
**Recommendation:** 6

**Clarity, Quality, Novelty And Reproducibility:**

The paper is well written and organized except for a couple of typos. The idea is a bit novel. It would be better to release the source code.

**Strength And Weaknesses:**

* Strength
1. The idea of adding Boundary Connectivity (BCXN) is interesting and novel
2. The proposed loss function can be inserted into any deep neural network architecture
3. Extensive experimental results show that the introduced loss function can significantly improve accuracy while reducing training iterations.

* Limitations
1. Compared to existing work on PINN. There exist many studies on PINN, so it would be better to discuss related work and compare the proposed method with some latest baselines, such as [Zeng et al 2022]
2. There is a typo about AQ = AP+ AQ in the first line on Page 6.
3. Question:
    I am curious if the proposed method can solve the prey and predator system which is not stable?

References:
[Zeng et al 2022] Competitive Physics Informed Networks, https://arxiv.org/pdf/2204.11144.pdf

**Summary Of The Paper:**

This work proposed a new Boundary Connectivity (BCXN) loss function to improve the accuracy and convergence speed of PINN. Specifically, it developed two variants of BCXN loss, namely: 1) a soft forcing method which imposes a linear approximation constraint via an additional loss term, and ii) a direct forcing approach which strongly enforces a linear constraint during the evaluation of PDE loss at nearboundary samples. Experimental results on multiple datasets demonstrated that the proposed method can significantly improve accuracy while reducing training iterations.

**Summary Of The Review:**

The contributions of this work lie in adding Boundary Connectivity (BCXN) in the loss function, which can improve accuracy and reduce training iterations. In addition, extensive experiments are conducted to validate the effectiveness of the BCXN loss function. The idea is a bit novel and interesting,

---

> ### Author Response · Authors · 2022-11-19
> **Authors Response**
>
> We thank the reviewer for the careful reading of our manuscript and comment on the limitations.
>
> > Compared to existing work on PINN. There exist many studies on PINN, so it would be better to discuss related work and compare the proposed method with some latest baselines, such as [Zeng et al 2022]
>
> We chose to compare different variants of the PDE loss as a baseline in this work and have summarized the results accordingly in **Appendix A18**. In the revised manuscript, we also noted that current proposed method can be jointly implemented with other PINN advances such as in loss balancing, domain decomposition, adaptive sampling, and other improved optimization methods such as [Zeng et al 2022], and that those approaches are not exclusive. Hence, for the purposes of this work, we believe it would be beneficial to focus on showing the intuition and results in isolation relative to our chosen baseline.
>
> > There is a typo about AQ = AP+ AQ in the first line on Page 6.
>
> We thank the reviewer for the careful reading of the manuscript and apologize for the error. We have updated this in the revised manuscript.
>
> > Question: I am curious if the proposed method can solve the prey and predator system which is not stable?
>
> We believe that the proposed methodology can solve the potentially unstable systems like the predator and prey system. In particular, as long as the problem can be described within a computational domain with any Dirichlet, Neumann or Robin-type boundary condition as detailed in our newly added **Appendix A1**, this method is applicable. However, we are unsure if the proposed BCXN loss will show any significant advantage in problems utilizing periodic boundary conditions (as might appear in some predator and prey model problems) as the points outside of the geometry can be adequately represented by utilization of the periodic condition.
>
> In addition, we have added results for an unsteady problem (in **Appendix A15**) that illustrates the improvement from this proposed method on the non-linear dynamical evolution of a fluid problem. This also matches one’s intuition that improved learning of initial time (or boundary) from the proposed method is important to the proper capture of non-linear, unstable dynamical evolution in systems. Nonetheless, we similarly agree with the reviewer that it will be interesting to explore (in future extensions) the limits of the proposed method (and PINNs in general) when learning potentially unstable physical systems.
>
> > The paper is well written and organized except for a couple of typos. The idea is a bit novel. It would be better to release the source code.
>
> Due to the concerns about maintaining author anonymity, we have chosen to include the relevant github links in the later versions of this manuscript. However, we will make our source code freely available as soon as possible in order to facilitate open collaboration and scientific exchange.

---

> > ### Comment · Reviewer_3A2P · 2022-11-20
> > **Thank you for your rebuttal**
> >
> > Thank you for your rebuttal! My concern is that PINN fails to solve the unstable systems, such as prey and predator,  since I found this issue in my own experiments. I hope the author may discuss this limitation and leave it to future work.

---

> > > ### Author Response · Authors · 2022-11-25
> > > **Thank you for your comment**
> > >
> > > Indeed, solving the unstable system is quite a challenge in PINN. From our experience, a very small violation from PDE constraints early in time solution can propagate and accumulate in later time, and snowball into much larger error, which is especially detrimental in unstable systems. Our current approach doesn’t mitigate this in a direct way, although a slightly more accurate solution (potentially from the BCXN concept) may indirectly help.
> > >
> > > We noted your further comment on discussing this limitation, and to leave it as our future work. We will insert this discussion into the manuscript in the next revision.

---

### Official Review · Reviewer_DAxW · 2022-10-28

**Confidence:** 4
**Correctness:** 4
**Technical Novelty And Significance:** 3
**Empirical Novelty And Significance:** 2
**Recommendation:** 5

**Clarity, Quality, Novelty And Reproducibility:**

The claims of the paper are clear and easy to follow.  The whole structure of the paper is good and well-written.
The work proposes a novel method for training PINNs derived from numerical computing works which is a great idea since these works are sufficiently mature since several decades.
The supplementary part contains all the necessary ingredients to reproduce the experiments. However, it would be better if the code could be made available.

**Strength And Weaknesses:**

In this work, the proposed method makes the training of PINNs feasible and sufficiently accurate at the boundary layer in a sparse sample regime where existing works fail (or require a highly dense grid at the boundary layer which is not easy to construct and/or is intractable in practice). Moreover, the method shows an extra gain in terms of computational complexity and achieves accurate solutions with drastically fewer iterations in a scarce data regime.
At the (near) boundary layer, the fluctuations are highly complex and the geometries are irregular. The proposed design of fast-PINNs loss allows in some way to cope with that. It is challenging because, given the local complexity near the boundary layer, sparse sampling leads to overfitting even though PDE constraint is applied to all the samples.

However, l have some considerations:

1/ It is important to underline that the success of PINNs relies on the successful propagation of information from initial and boundary condition points to interior points. This can be observed if we reduce the PDE solving to a local part of the domain. Do you think that your model achieves this successful propagation? how can you quantify that empirically?

2/ In existing PINNs works, the propagation failures are tied to highly imbalanced PDE residual fields.  In other terms, residuals with high values could be observed over very narrow regions. Does the proposed work mitigate/circumvents this failure?  if yes, how does the method succeed to make the balance of PDE residual fields?

3/ Given your method and the impact of a good choice of sampling strategy to make PINNs successful. I am wondering if the breakthrough work on PINNs is more on finding an appropriate sampling strategy near the boundary layer. If the latter is found, standard PINNs loss will work. What do you think?
A kind of sampling that focuses on points near the boundary layer with high PDE residual at every iteration. In practice, one could imagine sampling more of the points with high residuals.

4/ What do you think of adding the diffusion term in the loss function?

5/  Why did you set manually the different weights of each loss term rather than learning them?

6/ You mentioned that “ It is also very challenging to locally refine the sampling density for PINNs with CNN-architecture”. Recently, lots of efforts have been made to generalize learning to non-Euclidean data, known as Geometric Deep Learning. The latter is suitable to operate on meshes and irregular geometries, namely Graph Neural Networks (GNNs), PointNet.  Are there any reasons for not using GNNs ? Several works on ML&physics rely on these architectures.

7/ Lack of quantitative results. This work could be improved by adding a table of baseline w.r.t existing works on PINNs that proposes different variants of physical loss and w.r.t. different. sampling strategies

8/ It would be interesting to see what the proposed method could bring if it is combined with a given domain decomposition method.

9/ This work could be extended to more complex geometries such as the ones available in the different datasets proposed in the work entitled "Learning to simulate complex physics with graph networks". It allows to asses the generalization capabilities of the proposed method.

10/ Do you think that the proposed method could generalize well to high Reynolds (> 10^6)

11/ The figures could be updated with log scales

**Summary Of The Paper:**

The paper proposes a novel design of physical loss to find a trade-off between sample density and solution accuracy of PDEs. The formulation of the proposed loss tackles the challenges of approximating the boundary layer by modeling and adding a term in the loss function that takes into consideration the highly complex fluctuations of the gradients near the boundary layer. The work is inspired by a method in numerical computing.


**Summary Of The Review:**

The contribution of the proposed work will help to make progress in ML&Physics and build efficient PINNs.
I can increase my score depending on the responses.

---

> ### Author Response · Authors · 2022-11-19
> **Authors Response to Comments 1-3**
>
> > 1/ It is important to underline that the success of PINNs relies on the successful propagation of information from initial and boundary condition points to interior points. This can be observed if we reduce the PDE solving to a local part of the domain. Do you think that your model achieves this successful propagation? how can you quantify that empirically?
>
> Thanks for highlighting this point. We agree with the intuition that the success of PINNs relies on the successful propagation of information from boundary condition (and initial condition) points to the interior (and later in time points), with the latter also nicely covered in work by [1, 2]. This is one of the key motivations for this work, as we note that improper learning at the boundaries in more complex geometries greatly impacts accuracy of solution for boundary value problems. In addition, we note that there is often a compromise between perfectly matching the boundary/initial conditions and the ease of propagation. We believe that this work describes a potential avenue to accomplishing this via a loss function that provides structure for local linkage between the boundary points and the interior points, thereby facilitating the successful propagation of information.
>
> We plotted a contour of the local PDE loss function values across sample points in the domain. Indeed, when PINN fails, the PDE loss appears abnormally small inside the domain relative to values nearer the boundary. This is contrary to our observation with the inclusion of BCXN-loss, and suggests that the enhanced connectivity from our method helps to sustain information propagation from the boundary during training. This additional discussion has been included in the revised manuscript in **Appendix A8**.
>
> 1. Wang et al. "Respecting causality is all you need for training physics-informed neural networks."  (2022).
> 2. Daw et al. "Rethinking the importance of sampling in physics-informed neural networks." (2022).
>
> > 2/ In existing PINNs works, the propagation failures are tied to highly imbalanced PDE residual fields. In other terms, residuals with high values could be observed over very narrow regions. Does the proposed work mitigate/circumvents this failure? if yes, how does the method succeed to make the balance of PDE residual fields?
>
> As described above, we believe that this proposed work is indeed an alternative to mitigating inability to propagate information. The proposed loss function provides local linkage between boundary points and interior points via the lower order approximation, facilitating sustained information propagation from the boundary regions into the domain, and mitigating the effect of highly imbalanced PDE residuals. More discussion has been included in the revised manuscript in **Appendix A8**.
>
> > 3/ Given your method and the impact of a good choice of sampling strategy to make PINNs successful. I am wondering if the breakthrough work on PINNs is more on finding an appropriate sampling strategy near the boundary layer. If the latter is found, standard PINNs loss will work. What do you think? A kind of sampling that focuses on points near the boundary layer with high PDE residual at every iteration. In practice, one could imagine sampling more of the points with high residuals.
>
> We agree with the point that a sampling-based strategy can also lead to significant breakthroughs. This has been a fascinating on-going direction of work as detailed separately in our **Related Work** section with many possibilities in implementation. However, while adaptive (re)-sampling can be helpful, we believe this strategy also entails further development to address other practical challenges, including potential computational cost for computing re-sampling, (re)-balancing of the loss function as high/low residual points are chosen, etc. In addition, we observed that simply increasing sampling points in high residual regions can adversely affect training depending on factors such as local complexity and stiffness. We therefore envision that multiple complementary strategies could all be helpful for advancing PINN training and the current proposed method is suitable for co-implementation with other sophisticated sampling strategies to maximise potential benefits. While this would be an interesting direction, for the purposes of this work, we believe it beneficial to focus on showing the intuition and results in isolation. As a last point, we note that adaptive or residual-based sampling might be more intuitively suited to MLP- relative to CNN-type PINN implementations. On the other hand, the proposed method in this work has been shown to be beneficial and amenable for different types of PINN architecture. Nonetheless, we have brought some of this discussion into the revised manuscript, especially with regards to how we believe this method can be complementary to concurrent developments in sampling strategies.

---

> ### Author Response · Authors · 2022-11-19
> **Authors Response to Comments 4-8**
>
> > 4/ What do you think of adding the diffusion term in the loss function?
>
> Based on intuition from numerical methods, it is possible that the addition of diffusion into the loss function may make PINNs easier to train, however, this may also come at a slight cost in terms of accuracy of the solution and may even be harmful in certain regions. Properly balancing this trade-off may require more domain expertise and understanding of the specific physics of the problem being investigated. The proposed method here infuses the boundary condition into the training loss via a lower order approximation of known form which is then propagated to the rest of the domain. This however avoids any further unwanted deterioration in the internal domain by the addition of diffusion.
>
> > 5/ Why did you set manually the different weights of each loss term rather than learning them?
>
> We agree with the reviewer that there are other sophisticated methods in the literature to adaptively modify the loss weights [1-4]. In this particular demonstration, we note that the addition of the BCXN-loss term may further result in very different weight trajectories during PINN training, which complicates comparison of results. Hence, based on our own prior experiences in earlier work, we refined the weights for better performance and used the manual setting for consistency in comparison. We believe this would be an additional interesting direction to explore in future work though.
>
> 1. Elhamod et al. "CoPhy-PGNN: Learning physics-guided neural networks with competing loss functions for solving eigenvalue problems." (2020).
> 2. Wang et al. "Understanding and mitigating gradient flow pathologies in physics-informed neural networks." (2021).
> 3. Wang et al. "When and why PINNs fail to train: A neural tangent kernel perspective." (2022).
> 4. Jin, Xiaowei, et al. "NSFnets (Navier-Stokes flow nets): Physics-informed neural networks for the incompressible Navier-Stokes equations." (2021).
>
> > 6/ You mentioned that “ It is also very challenging to locally refine the sampling density for PINNs with CNN-architecture”. Recently, lots of efforts have been made to generalize learning to non-Euclidean data, known as Geometric Deep Learning. The latter is suitable to operate on meshes and irregular geometries, namely Graph Neural Networks (GNNs), PointNet. Are there any reasons for not using GNNs ? Several works on ML&physics rely on these architectures.
>
> We agree with the reviewer that recent developments in geometric deep learning have indeed been very exciting. Nonetheless, we have focused our development on MLP and CNN- type architectures as these are still most used in current state-of-the-art PINN work. For instances, CNN-type PINNs have certain advantages in their ease of representation and amenability to numerical-type ideas like Taylor series expansions. GNNs will be an interesting next step for demonstration, and we very much plan to explore this direction further, but additional developments will be required, just as unstructured meshes in numerical methods typically require some customized treatment relative to uniform and structured meshes.
>
> > 7/ Lack of quantitative results. This work could be improved by adding a table of baseline w.r.t existing works on PINNs that proposes different variants of physical loss and w.r.t. different. sampling strategies
>
> We thank the reviewer for highlighting this point. We did compare different variants of the PDE loss formulation and have summarized the results accordingly in **Appendix A18**. However, as described in the section above, the sampling strategies mentioned in many of the prior work tend to focus on MLP-type automatic differentiation-based PINNs. Hence, we believe that a comparison with regards to sampling strategies is not the best comparison for this work. For example, the CNN-type PINN mentioned in our lid-driven cavity meta-model is not amenable to variations in sampling strategies as it operates on a pre-defined voxel-type system. Nonetheless, as mentioned above, we have included some comparison to other baseline existing work on PINNs in a summary table in **Appendix A18**.
>
> > 8/ It would be interesting to see what the proposed method could bring if it is combined with a given domain decomposition method.
>
> We are grateful that the reviewer has suggested multiple potential additional extensions to the ideas proposed in this work. Indeed, we mean for this to be a first demonstration of the efficacy of the method and are similarly excited to explore potential extensions of this method in ways such as facilitating physically meaningful, less constrained domain decomposition for enhanced PINN modelling. This suggestion by the reviewer could bring some interesting benefits and results, but due to the current time and length constraints, we will explore this more in future work.

---

> ### Author Response · Authors · 2022-11-19
> **Authors Response to Comments 9-11**
>
> > 9/ This work could be extended to more complex geometries such as the ones available in the different datasets proposed in the work entitled "Learning to simulate complex physics with graph networks". It allows to asses the generalization capabilities of the proposed method.
>
> We agree with the reviewer that the work mentioned does have other interesting geometries and physics. Some of them involved particle-based simulation methods and would involve further methodology development before we can test our proposed method which focuses on the BCXN-loss component. Nonetheless, we appreciate the suggestion, and have included additional results on flow past an airfoil which is one of the many interesting scenarios studied in work from that group. The results for this experiment appear consistent with the previously observed benefits from our proposed method, and are presented in more detail in **Appendix A6**. Nonetheless, we do acknowledge that room remains for further extensions towards more complicated, real-world problems.
>
> > 10/ Do you think that the proposed method could generalize well to high Reynolds (> 10^6)
>
> In theory, the proposed method is physics-agnostic, and the use of a linear-type constraint described for the BCXN-loss can be applied to low and high Reynolds problem for better modelling of the geometrical boundaries. However, although we have demonstrated advantages relative to the baseline PINN implementations, these problems are still on the order of Re~1000. Other important effects at higher Re can greatly impact PINN training feasibility, including proper representation of the boundary layer and turbulence effects (especially as our current PINN implementations do not incorporate any turbulence modelling). Nonetheless, we believe that the proposed method could be a means for incorporating wall-function-type representations into PINN model training (via the BCXN-loss), which has been shown to be useful for reducing computational cost in computational fluid dynamics, and this could in turn enable modelling of high Reynolds number.
>
> > 11/ The figures could be updated with log scales
>
> We agree with the reviewer that some of the absolute error plots can be better represented with log scales, and have ensured that our MSE plots are all on log scales in the revised manuscript. However, as field variables like velocity and pressure can span both positive and negative values, we choose to retain those on a linear scale for better representation. We also tried to plot the velocity magnitude contour in log scale, but the visual effect is not representative. Therefore, we chose to plot all the velocity magnitude contours in linear scale.
>
> > The claims of the paper are clear and easy to follow. The whole structure of the paper is good and well-written. The work proposes a novel method for training PINNs derived from numerical computing works which is a great idea since these works are sufficiently mature since several decades. The supplementary part contains all the necessary ingredients to reproduce the experiments. However, it would be better if the code could be made available.
>
> We thank the reviewer for the thoughtful suggestions for improvement and the many ideas for further extension. Indeed, we agree that numerical computing can be a great source of inspiration for novel methods to train PINNs better. Due to the concerns about maintaining author anonymity, we have chosen to include the relevant github links in the later versions of this manuscript. However, we will make our code freely available as soon as possible in order to facilitate open collaboration and scientific exchange.

---

### Decision · Program_Chairs · 2023-01-20

**Decision:**

Reject

**Justification For Why Not Higher Score:**

relatively minor contribution

**Justification For Why Not Lower Score:**

good experimental results

**Metareview: Summary, Strengths And Weaknesses:**

The paper proposes a way to handle boundary conditions for training physics-informed neural networks (PINNs) when the required derivatives are computed with numerical differentiation methods and the boundary condition points do not coincide with the collocation points. The proposed idea is simple and the experimental results are encouraging. The reviewers think that the work is quite well executed although the work would benefit from more extensive evaluation of the proposed method, for example, by considering other types of boundary conditions or by using more advanced interpolation techniques instead on the linear one. The reviewers think that Figure 2 is very misleading, it creates an impression that solving this equation is problematic for PINNs although many PINN papers report good results on this equation. We strongly recommend the authors to improve the description of Figure 2 and the accompanying claims. In conclusion, the novelty exists but it is not large, the execution and writing is good but could be improved. Because of the relatively small contribution and the high standards of ICLR, I recommend rejection.

**Summary Of Ac-Reviewer Meeting:**

During the meeting, the reviewers came to a general agreement that although the idea of the paper is largely borrowed from the numerical methods literature, a combination of this idea with PINNs is an interesting contribution. The reviewers think that the work is quite well executed although the work would benefit from more extensive evaluation of the proposed method, for example, by considering other types of boundary conditions or by using more advanced interpolation techniques instead on the linear one. The reviewers think that Figure 2 is very misleading, it creates an impression that solving this equation is problematic for PINNs although many PINN papers report good results on this equation. We strongly recommend the authors to improve the description of Figure 2 and the accompanying claims.